psychology

preregistration, researcher degrees of freedom, open science, psychological science, transparency

**Author for correspondence:**
Aline Claesen
e-mail: aline.claesen@kuleuven.be

# Comparing dream to reality: an assessment of adherence of the first generation of preregistered studies

Aline Claesen, Sara Gomes, Francis Tuerlinckx and Wolf Vanpaemel

Faculty of Psychology and Educational Sciences, KU Leuven, Tiensestraat 102, Leuven 3000, Belgium

 AC, 0000-0002-0303-2441; SG, 0000-0002-2099-6314;
FT, 0000-0002-1775-7654; WV, 0000-0002-5855-3885

Preregistration is a method to increase research transparency by documenting research decisions on a public, third-party repository prior to any influence by data. It is becoming increasingly popular in all subfields of psychology and beyond. Adherence to the preregistration plan may not always be feasible and even is not necessarily desirable, but without disclosure of deviations, readers who do not carefully consult the preregistration plan might get the incorrect impression that the study was exactly conducted and reported as planned. In this paper, we have investigated adherence and disclosure of deviations for all articles published with the Preregistered badge in *Psychological Science* between February 2015 and November 2017 and shared our findings with the corresponding authors for feedback. Two out of 27 preregistered studies contained no deviations from the preregistration plan. In one study, all deviations were disclosed. Nine studies disclosed none of the deviations. We mainly observed (un)disclosed deviations from the plan regarding the reported sample size, exclusion criteria and statistical analysis. This closer look at preregistrations of the first generation reveals possible hurdles for reporting preregistered studies and provides input for future reporting guidelines. We discuss the results and possible explanations, and provide recommendations for preregistered research.

## 1. Introduction

During the research process, researchers inevitably make numerous decisions, collectively known as researcher degrees of freedom [1]. Among others, researchers need to decide on

the number of participants, possible data transformations, the treatment of outlying data, the inclusion of covariates, and so on. When this flexibility is exploited, the probability of a type I error is drastically increased [2,3], or the effect size can be inflated. A common example is the practice of optional stopping, which involves stopping data collection based on interim data analysis, in order to reach the desired result. Moreover, even when researchers have no intention to actively fish for desirable results, an inspection of the data can still influence the way analyses are conducted, thereby creating a multiple comparisons problem [4]. For a long time, researchers were incentivized to report exploratory results as confirmatory [5], which has contributed to a literature littered with plausibly non-replicable results. In a large-scale replication effort, the average effect size estimate in the 100 replication studies was only half as large as the one in the original study, and the number of studies with statistically significant outcomes decreased from 97 to 36 [6]. Without knowing the extent to which researcher degrees of freedom have been exploited, it is challenging to correctly interpret a published finding [7]. To help readers evaluate the credibility of a finding, radical transparency has been proposed as a solution [8,9]. One way to increase research transparency is to preregister research, which involves freezing decisions regarding the study (e.g. study design, data collection and data analysis) on a public third-party repository prior to seeing, or ideally prior to collecting, the data [5]. By specifying decisions before data collection, researcher degrees of freedom are restricted, and decisions that are made during the data collection and analysis cannot be mistakenly reported as *a priori*. Therefore, the practice is most suitable for confirmatory research but does not favour it. It enables to distinguish confirmatory from exploratory results [5].

Preregistration has been strongly advocated in psychology (although there is also criticism, e.g. [10]). For example, several journals, such as *Psychological Science* [11], have adopted an incentive system that acknowledges preregistration in the form of badges provided by the Center for Open Science (hereafter, COS). Some journals, such as *Cortex* [12] have taken preregistration as the cornerstone of a paper format, called *registered report*, in which the preregistration plan is reviewed before the study is conducted, and accepted studies are guaranteed to be published after data collection. Preregistration plans are not prisons [13]. After having collected the data, researchers may need to divert from their preregistration plan. Deviations can occur for a variety of reasons, such as for the sake of scientific discovery, non-feasibility of the preregistered plan because of violation of model assumptions, unexpected circumstances, errors in the preregistration plan, suggestions from reviewers, new insights, or it can happen inadvertently [14]. If these deviations are not clearly disclosed in the manuscript, a reader might get the false impression that the study was preregistered exactly as reported.

In the field of clinical trials, where prospective registration is already prevailing [15], discrepancies between registered and reported outcomes, often referred to as outcome switching, are prevalent (e.g. [16–20]). In the current study, we focus on deviations from the preregistration plan in pioneer research articles with a *Preregistered badge* in *Psychological Science*, and whether these deviations are disclosed. We identify possible explanations for deviations and formulate several recommendations to increase transparency in preregistered studies in future research.

# 2. Methods

## 2.1. Disclosure

We report how we determined our sample size, all data exclusions and all measures in the study. We opted for an exploratory approach, and we did not preregister any *a priori* choices. Part of our measures and procedure were influenced by what we encountered in the preregistration plans and the articles. First of all, it was only after being confronted with the large variations in the accessibility of preregistration plans and in the number of details provided that we implemented two exclusion criteria for the adherence assessment, as discussed below. Also, adherence was initially assessed using eight items rather than the final six. We removed two items, due to inconsistencies in the interpretation of these items.[1] Further, the decision to contact the corresponding authors of the preregistered studies was made after our assessment, following a suggestion from a reviewer of a previous version of this paper. Finally, we initially planned to keep track of the timing of preregistration (during or after data collection, as

---

[1]A previous version of this article, still available as a preprint on https://psyarxiv.com/d8wex/, reports the assessment with eight items, before the vetting by the corresponding authors. We incorporated the *direction of effect* in the first item, the *research question and hypothesis*, and *operationalization* of the variables in the second item, the *variables*.

indicated by the authors), the type of study (new or replication), and the compliance with the open practices disclosure items[2] as well, but that turned out to be infeasible.[3] Our coding scheme and assessment can be found here: https://osf.io/f49an/. Analyses are conducted in R (v. 4.0.2), employing the following R packages: *xlsx* (v. 0.6.5), *tidyverse* (v. 1.3.0), *ggplot2* (v. 3.3.2) [21–24]. R script for main outcomes and figures 2 and 3 are available here: https://osf.io/bs5q2/.

## 2.2. Sample

We collected all articles that were published with a Preregistered badge in *Psychological Science*, between February 2015, the issue containing the first article with a Preregistered badge, and November 2017, which was the start date of this study. The motivation for this approach was that the Preregistered badge was a straightforward way to detect published preregistered studies. On top of that, *Psychological Science* was one of the pioneer journals that encourage transparency with, among other initiatives, the implementation of the incentive system provided by the COS [11]. The Preregistered badge indicated the presence of a preregistration plan. In order to earn the badge, authors had to fill out an open practices disclosure document, in which they declared that there is a permanent path to a preregistration plan on an online open access repository, and in which they could disclose deviations if any. The preregistration plans were not reviewed. Our selection resulted in 23 articles (four from 2015, four from 2016 and 15 from 2017) and 38 preregistration plans. There were more preregistration plans than articles in our sample because seven articles contained multiple studies that were preregistered separately. For one article, an erratum was published in 2020, and we included the corrected version in our sample rather than the original version published in 2017.

To facilitate the comparison between the preregistration plan and the study as described in the published article, the plan needed to satisfy two requirements: it should be easily accessible, and it should contain a minimal number of methodological details. First, accessibility is necessary because if it was not clear where to find the document that acts as the preregistration plan, checking adherence to the plan was impossible. Second, we required certain methodological details to be preregistered in order to be able to assess adherence for the same details in each study. That is, the purpose was to select studies of which adherence could be evaluated regarding six methodological items that indicated confirmatory hypothesis testing; the minimal detail criterion did not constitute an evaluation of the quality of the preregistration plan. Application of both exclusion criteria resulted in a selection of 16 articles and 27 corresponding preregistrations (see Results for more details about the excluded preregistrations). Note that we only excluded these studies from the adherence assessment, but not from secondary outcomes.

## 2.3. Measures and outcomes

Both exclusion criteria for the adherence assessment, accessibility and minimal methodological detail, were assessed using six items each that were scored with either 0 (unfulfilled) or 1 (fulfilled). The items for accessibility were the following: a preregistration plan should be a (i) permanent, (ii) read-only, (iii) time-stamped, (iv) public, (v) non-ambiguously available, and (vi) on a third-party repository stored document. If not all items were scored with a 1, the plan was excluded. Minimal detail was assessed for the remaining preregistration plans by evaluating whether they contained some information on the following methodological aspects: (i) a research question and/or hypothesis, (ii) a list of variables, (iii) operationalization of the variables, (iv) a sample size, (v) a procedure, and (vi) a statistical model. Again, if not all items were present, the plan was excluded. We did not assess the degree of detail per item.

The main outcome measure in our study is the adherence of the study as reported in the published article to the preregistration plan. We compared the reported study and the preregistration plan on six

---

[2]Every article that was published in *Psychological Science* with one or more open practices badges was accompanied by an open practices disclosure document, containing five disclosure items adopted from the guidelines of the OSF (https://osf.io/tvyxz/wiki/2.%20Awarding%20Badges/).

[3]We left out these measurements because firstly, only a few authors reported in the preregistration plan or on the OSF registration form when they preregistered relatively to data collection. On top of that, it is hardly possible to verify the statements of the authors. Secondly, an unambiguous distinction between replication or no replication did not seem feasible. Finally, compliance to the open practices disclosure items was assessed; however, because this assessment highly depends on the interpretation of the items and because it was done by only one rater, we do not report this assessment.

items: (i) research question and/or hypothesis, (ii) list of variables, (iii) sample size, (iv) exclusion criteria, (v) procedure, and (vi) analysis. Each item was coded with one of the following three options: *no deviations*, *undisclosed deviation(s)* or *all deviations disclosed*. Note that the adherence items somewhat differed from the minimal detail items. In particular, we selected studies that listed variables and their operationalization (i.e. what and how the variables would measure or control). Due to frequent changes in terminology, we sometimes had to identify variables based on their description. Therefore, the variables item in the adherence assessment covers operationalization as well. Also note that we did not require exclusion criteria for minimal detail, but did include this item in the adherence assessment. If no exclusion criteria were reported in the paper, and no exclusion criteria were preregistered, then there was no deviation.

If there was no clear discrepancy between the preregistration plan and the study as reported in the article, then the item was coded as *no deviations*. When there was a clear discrepancy between what is preregistered and what is reported in the published paper, we looked for disclosure in the main text, in the notes, in the electronic supplementary material, in the open practices disclosure and, if referred to in one of the former, on the Open Science Framework (hereafter, OSF). An item was coded as *all deviations disclosed* only when there was full disclosure. For example, if authors reported two non-preregistered exclusion criteria but only disclosed one of these two as a deviation from the plan, we did not consider this as full disclosure, but as *undisclosed deviation(s)*. Parts of the papers that were clearly labelled as exploratory were not included in the comparison (i.e. this was coded as no deviations rather than all deviations disclosed).

Besides our main outcome (adherence), we also kept track of two secondary outcomes for all preregistration plans: template use and repository for all preregistrations. First, for template use, we classified each preregistration plan as using no template or as using a template. If authors used a template, we indicated which one. We compared the structure of the preregistration plans with the templates we found in 2017: the format from Aspredicted.org, the template suggested by van 't Veer & Giner-Sorolla [25], the replication recipe suggested by Brandt *et al.* [26] and the COS prereg challenge registration form (now called OSF preregistration, see https://osf.io/prereg/). Second, we kept track of the repositories used to store the plans (e.g. osf.io, aspredicted.org, and clinicaltrials.gov). If the preregistration was stored on the OSF, then we specified whether this was on the wiki, the registration form or an uploaded file.

## 2.4. Procedure

Accessibility and minimal detail, and adherence, were assessed by two independent raters—the first and second authors of this paper. After a first independent assessment, the raters discussed their findings for every item until they reached a consensus whenever there was a disagreement. Accessibility was rated for all papers, minimal methodological detail was rated for all accessible papers, and finally, adherence was assessed for all studies that passed the exclusion criteria (figure 1). Upon some changes to the coding scheme (see Disclosure section), the adherence assessment was updated by one rater, under the supervision of the two last co-authors (i.e. every change was discussed). Assessing the adherence of the published studies to the preregistration plans proved to be a far from trivial task. The main obstacle was that often neither the preregistration plans nor the published studies were written in sufficient detail for a fair comparison. As a result, there were cases in which it was difficult to assess whether there was a deviation from the preregistration plan. Whenever there was reasonable doubt about a deviation from the preregistration plan (i.e. the plan and the paper were consistent despite varying degrees of specificity), this was coded as no deviations. The assessments were shared with the corresponding authors of the papers in the sample, which lead to some changes in the assessment. Comments from corresponding authors, changes in our initial assessment, and remaining disagreements are discussed in the Results section.

# 3. Results

## 3.1. Accessibility and minimal detail

A meaningful comparison of the preregistration plan with what is reported in the paper is only feasible when the plan is fairly easy to access and contains the minimal methodological details we required for the adherence check. In our sample, 11 plans (from eight papers) were excluded. Four plans were

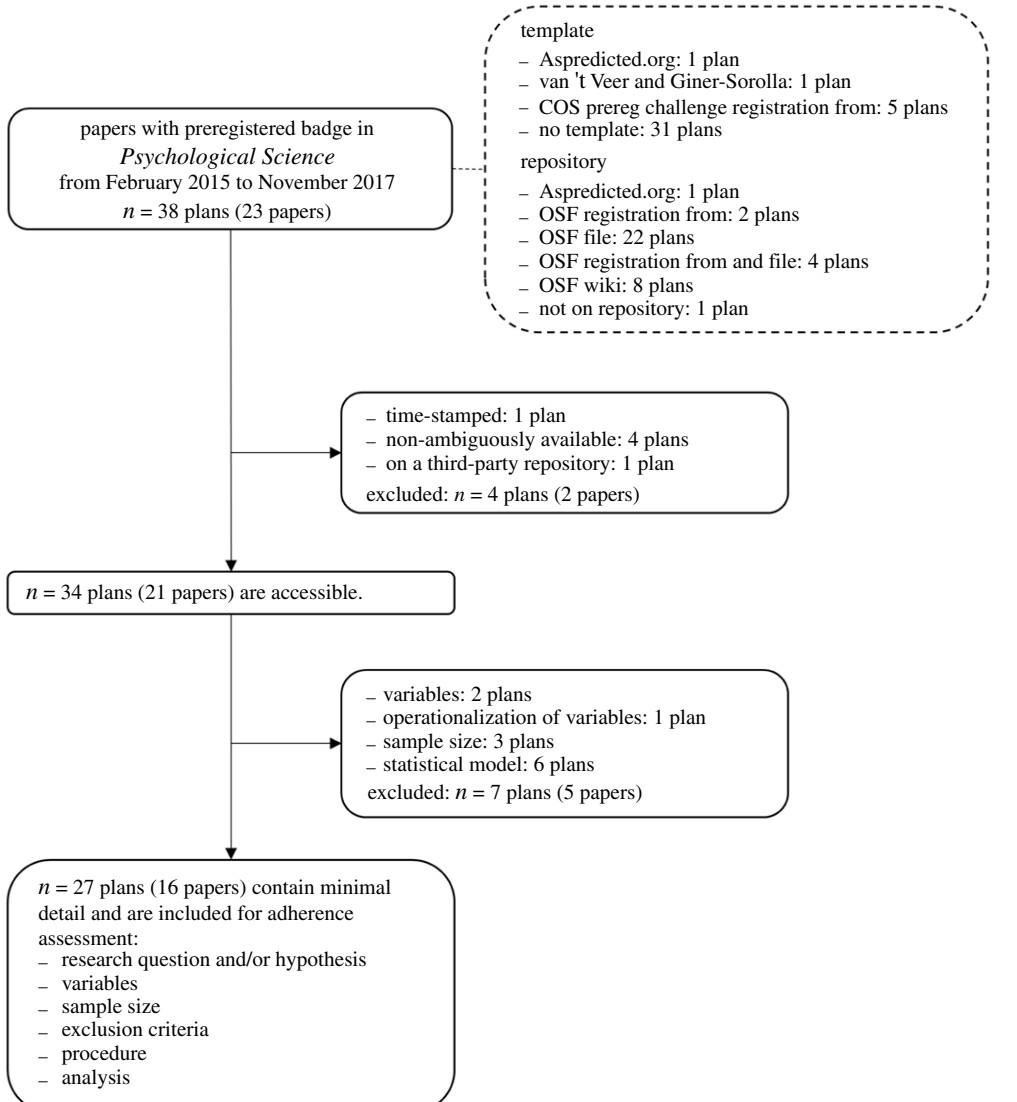

**Figure 1.** Flowchart of assessment of preregistered studies.

excluded because they were not unambiguously accessible, in the sense that it is not sufficiently clear what the exact final plan of interest was. One plan was difficult to find because the URL led to a non-frozen project on the OSF with multiple studies and registrations. On the frozen page, there were files with the wrong study number or 'study X'. The incorrect assumption that one of the files of 'study X' contains the preregistration plan for the study in question is easily made. For two studies from one paper, it was impossible to reconstruct the final preregistration plan with reasonable certainty, because there were different versions and partial updates, and a lack of structure. Finally, one plan was reported to be available on the OSF, but we did not find the latest version. However, the latest version was included as electronic supplementary material. As far as we are aware, on top of being ambiguously available, this plan was not time-stamped.

Seven plans were excluded because they did not contain the minimal methodological details. All excluded plans contained a research question or hypothesis and a procedure. However, six plans did not include statistical analysis, two did not list variables, one did not operationalize the variables, and three did not include a planned sample size.

## 3.2. Preliminary observations

One obstacle for the assessment of adherence was that authors regularly switched terminology between plan and paper. Terminology switches occurred in both the description of the statistical analysis (e.g.

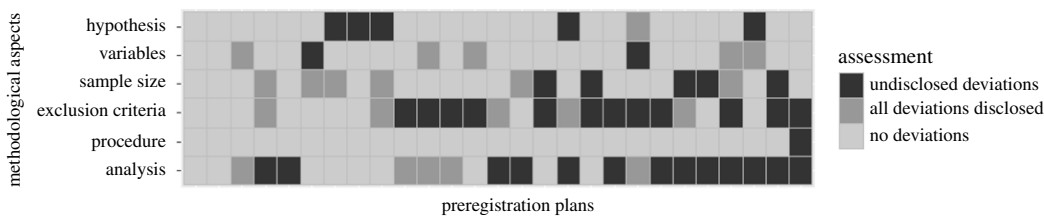

**Figure 2.** Tile plot of the assessment of each methodological aspect per preregistration plan. Only the 27 studies that were accessible and included the minimal number of methodological details required for our adherence assessment are shown.

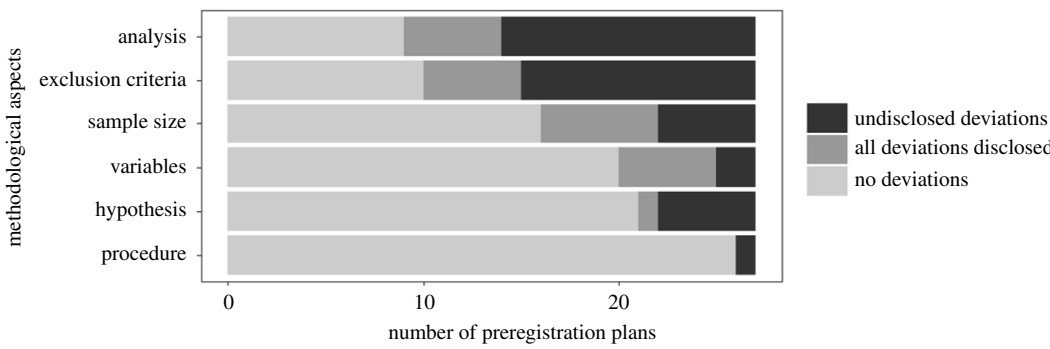

**Figure 3.** An overview of adherence per methodological aspect.

'one-sample *t*-tests' versus 'paired *t*-tests'), and in the variables (e.g. 'racial prejudice' versus 'racial bias', or 'located in the US' versus 'US citizen'). Similarly, some methodological details were presented differently. For example, 'the sample will consist of US citizens' versus 'non-US citizens will be excluded from the analysis'. Another complicating factor was that a few preregistration plans possibly contained typos and copy-paste errors, for example, 'left-skewed' versus 'right-skewed', and, in case of multiple studies, preregistering the same analysis twice instead of a new analysis. Because typos and errors can be disclosed, this was also coded as a deviation. Finally, there often was a difference in the level of specificity between the preregistration plan and the reported study. For example, in one study, the authors preregistered a regression model with possible predictors and interaction terms, but not what the model would exactly look like. In the article, the authors reported the model in more precise and definite terms. Although the model in the preregistration plan was not identical to the model reported in the manuscript, it was compatible with it. In cases like this, we did not code it as a deviation from the preregistration.

## 3.3. Main outcome: adherence

In our sample, two of the 27 (7%) selected studies did not deviate from the preregistered plan in any of the preregistered methodological aspects (figure 2). One study (4%) reported all the deviations. In the remaining 24 of 27 (89%) studies, there was at least one item for which a discrepancy between the preregistration plan and the journal article was not fully disclosed. None of the deviations was fully disclosed in the article for nine out of 25 (36%) preregistered studies with deviations. In the remaining 16 (64%) studies, at least one of the deviations were fully disclosed in the article.

Figure 3 provides an overview of adherence and disclosure per item. Non-adherence was most prevalent for the planned sample size, exclusion criteria and analysis. For six out of 27 (22%) studies, the deviation in sample size is disclosed, as opposed to five (19%) studies, where the deviation is left undisclosed. For five (19%) studies, all changes in exclusion criteria are reported, while in 12 (44%) other studies, this is not the case. Finally, in five (19%) studies, all deviations in the statistical analysis are reported, but in 13 (48%) not. For this reason, we will discuss these three aspects in more detail below.

### 3.3.1. Sample size

Usually, an exact or minimum sample size was preregistered, and in case of a deviation, there was often only a small discrepancy between the preregistered and the reported sample size. In seven studies, the

reported sample size was larger than preregistered, which was disclosed in five studies. Two studies reported a smaller sample size than preregistered, of which one was disclosed as a deviation. Finally, there were two studies in which the reported sample size was larger and smaller than planned, respectively, depending on data exclusion. Disclosure consisted of reporting both the preregistered sample size and the actual sample size, as in the following example. The authors preregistered:

'Our target sample size is 80. We will continue recruitment until we receive complete data from 80 children who pass the manipulation check […]. Data collection will be terminated once the target N of 80 is reached.'

The stopping rule was clear: data collection would stop once 80 children passed the manipulation check. However, not everything went as planned. In the article, the authors reported a sample size of 81 children. Although this is a small deviation from the preregistration plan, the authors disclosed it:

'Our predetermined target sample size was 80 participants. We mistakenly recruited an extra child into one of the counterbalancing conditions, which resulted in a sample size of 81.'

### 3.3.2. Exclusion criteria

Regarding the exclusion criteria, two kinds of deviations occurred. On the one hand, 12 studies reported one or more non-preregistered exclusion criteria. In four studies, this deviation was fully disclosed by distinguishing between preregistered and non-preregistered exclusion criteria. For example, one study reported:

'Participants who answered the manipulation check incorrectly ($n = 8$) were excluded from the analyses, as specified by our preregistered exclusion criterion. Four additional participants clearly did not follow task instructions, as they reported outcomes higher than 6 in more than 50% of the trials. They, too, were excluded. Thus, all analyses were conducted on the data from 48 participants (4,608 observations). Including the 4 participants whose exclusion was not based on our preregistered exclusion criterion did not change any of the results reported here.'

On the other hand, eight studies did not report at least one preregistered exclusion criterion. This was fully disclosed in one study, the authors reported in a footnote that not all exclusion criteria were reported because they were irrelevant and included where they could be found.

### 3.3.3. Statistical analysis

Deviations from the analysis plan were diverse. First of all, 14 studies reported non-preregistered analyses, such as an extra simple effect. Three studies disclosed all non-preregistered analyses. Conversely, five studies did not report preregistered analyses, for example, a robustness check. This kind of deviation was disclosed in two studies. Finally, seven studies reported a different model than planned. Four studies disclosed this deviation, such the following example:

'We reported two separate ANOVAS instead of a single MANOVA. While our original plan was to conduct a single MANOVA to compare the forecasts and peer rank estimates across targets, we chose instead to conduct two separate ANOVAS. We did this for two reasons: (i) because the two dependent variables were not highly correlated, which renders the approaches equivalent; and (ii) because conducting two separate ANOVAs allowed us to compare the slopes of the lines between the two measurements (i.e. comparing the linear trend of forecasts across targets with the linear trend of peer rank estimates across targets). Comparing the linear trends proved a simpler, more direct way to test our two competing hypotheses.'

## 3.4. Comments from corresponding authors

We asked all 23 corresponding authors of the 38 studies to comment on and, if necessary, correct our assessment. Of the contacted corresponding authors, 19 responded and provided feedback. The authors of the remaining four papers (which were all excluded from the adherence assessment) could not be reached or did not wish to comment. Regarding our exclusion criteria, accessibility and minimal detail, no assessments were changed. The authors provided us with valuable notes on why our criteria were not met. In particular, for the minimal detail criterion, two corresponding authors pointed out that they did not preregister certain study details because of the nature of their study. One conducted a direct replication and indicated that study details could be retrieved from the original study. Another corresponding author clarified that it was not realistic to set the sample size beforehand for their observational study.

Regarding the primary outcome, adherence, we changed 14 out of 162 (9%) assessments based on the feedback from corresponding authors. Eleven assessments were switched from 'undisclosed deviations' to 'no deviations' in 10 studies across six papers. Three assessments in two studies from the same paper

(two regarding the hypothesis and one regarding the analysis) were changed, because the authors showed where in the paper we could find the details that we classified as not reported. One assessment regarding the hypothesis was changed, because we misread the preregistration plan. We changed one assessment regarding the procedure, because the authors argued the deviation was too arbitrary, and we agreed. In three studies from two papers, we changed our assessment regarding the sample size, because of a misinterpretation of the stopping rule on our behalf. Finally, three switches concerned the assessment of the analysis in three studies from the same paper. The authors noticed that the deviation was actually in the hypothesis and not in the analysis. Therefore, three assessments regarding the analysis changed from *undisclosed deviations* to *no deviations*, and three assessments regarding the hypothesis in these studies were switched from *no deviations* to *undisclosed deviations*.

Our correspondence with the authors provided some insight into the genesis of the deviations. They clarified that they deviated from the preregistration plan, for example based on suggestions from reviewers. Another common reason for non-disclosure was that there were no actual deviations in conducting the study, but that authors simply did either not preregister or not report certain details because they were redundant or irrelevant. For example, several corresponding authors argued that removing duplicate IP addresses was considered standard practice, and therefore either not preregistered or not reported in the paper. Because there was no consensus on what should be disclosed, this sometimes led to a disagreement between our assessment and the corresponding author's assessment. In total, there were disagreements in eight (5%) coding decisions, which occurred in seven studies across four papers. Six disagreements in three papers concerned the sample size. In five studies, the authors acknowledged that they could have better described the planned sample size in the preregistration plan but argue the discrepancy with the preregistered sample size was too small to be considered a deviation. In one study, the corresponding author clarifies that platforms for online participant recruitment always sample slightly more participants than planned. However, it was not determined in the preregistration plan that the platform would be used to stop data collection. Finally, there were two disagreements regarding the statistical analysis in two papers.

## 3.5. Secondary outcomes

Seven out of 38 (18%) preregistration plans were constructed through a template. One was registered with the format from Aspredicted.org, one used the template suggested by van 't Veer and Giner-Sorolla [25], and five registered with the COS prereg challenge registration form. Three studies (published in the same article) out of those five also had a second version that did not follow the template. Both versions contained the same information, but the authors preferred their structure over the template. None of the plans that used a template was excluded. Note that templates were not available to all authors at the time of constructing the preregistration plan.

One preregistration plan was stored on Aspredicted.org. One was reported as stored on the OSF, but not found by us. The 36 remaining plans were stored on the OSF and found by us. Six were uploaded in an OSF registration form: five in the COS prereg challenge registration format and one in the open-ended format. Twenty-two plans were uploaded on the OSF in a file. Eight plans, of which five correspond with five studies reported in one article, were registered using the OSF wiki.

## 4. Discussion

In our sample of 23 papers published in *Psychological Science*, we found that all but two out of the 27 selected preregistered studies contained deviations from the preregistration plan, and that one preregistered study fully disclosed all the deviations. Deviations from a preregistered plan do not constitute evidence of exploited researcher degrees of freedom. On the contrary, if the preregistered plan suffers from methodological deficiencies, strictly following the plan compromises the quality of the study and would be bad statistical judgement. Further, given the importance of exploration in scientific discovery, zealously sticking to the plan can impede scientific progress [25]. However, disclosing all deviations from the plan is crucial for preregistration to improve transparency. Without carefully comparing the published studies to the preregistration plan, readers of these papers might be left with the wrong impression that these studies were exactly performed and reported as planned. Our preliminary observations also showed that even with the preregistration plan at hand, it is challenging to distinguish what was planned *a priori* from what was not. On top of that, not all preregistration plans were unambiguously accessible.

Our findings are consistent with those from other adherence assessments of preregistered studies. In the fields of economics and political science, Ofosu & Posner [27] found in 93 pre-analysis plans registered between 2011 and 2016 that over a third of the papers did not adhere to the planned hypothesis and 18% contained non-preregistered hypothesis tests, which were not disclosed in 82% of the cases. In a more recent study, Heirene *et al*. [28] reviewed a sample of 20 gambling studies preregistered between 2017 and 2020 and found that 65% contained undisclosed deviations. Corresponding authors of the preregistered studies in our sample indicated that deviations resulted from, among others, suggestions from reviewers, honest mistakes, such as typos in the preregistration plan, or that they simply considered certain details as too irrelevant to report either in the preregistration plan or in the paper. There are no standard guidelines on what should be disclosed in the paper, and opinions differ. Especially, the sample size and analysis plan were difficult to assess due to a lack of reporting standards for preregistered studies. For example, should researchers report that they deviated from the sample size when they only sampled a few participants less or more than planned? If the statistical model was reported in more detail in the paper in the preregistration plan, should researchers report these details as deviations?

## 4.1. Recommendations

Based on our observations, we make the following recommendations to improve preregistered research.

1. *Make the preregistration plan unambiguously accessible.* Accessibility can be improved by providing a direct link that brings the reader straight to the final preregistration document. Authors should try to avoid different versions of their plan, but when unavoidable, they should make clear what the differences between the versions are, and which version counts as the final preregistration plan. Further, the time stamp is more interpretable if authors report when data collection started.

2. *Make the preregistration plan sufficiently detailed*. As shown by Bakker *et al*. [29], it is difficult to freeze all researcher degrees of freedom by preregistration, and the interpretation can differ between readers. They used to be scarce, but meanwhile, there is a myriad of templates available for all kinds of study types. Following a template (e.g. https://osf.io/zab38/wiki/home/) or reporting standards adapted to the type of research conducted (e.g. the CONSORT statement for clinical trials; http://www.consort-statement.org/), can help with preregistering important *a priori* methodological details. Vague preregistration plans allow for too much wiggle room and do not contribute to transparency. Good practices include preregistering a stopping rule and making an explicit distinction between the sample size before and after excluding cases, and statistical models should be preregistered as specifically as possible.

3. *Review the preregistration plan*. The most efficient way to boost adherence, and the disclosure of deviations, involves reviewers of preregistered manuscripts evaluating adherence. This recommendation is beyond the control of authors but is part of the practice of the journals honouring preregistrations. An extra pair of eyes can not only inspect whether there are deviations but can also assess the interpretability of the study. For example, terminology switches can be detected by reviewers. For this reason, we suspect that a review-based approach, such as the format of registered reports [12], is superior to the disclosure approach in the sample of our study. However, Hardwicke & Ioannidis [30] found several implementation issues in registered reports as well, like non-availability of the plans that are in principle accepted and various ways of registering.

4. *Disclose deviations*. Few readers will take the time to meticulously compare the published paper with the preregistered plan. Instead, they will assume that, unless noted otherwise, the preregistered plan was followed and exactly as reported in the paper. It is a great service to the readers to disclose any deviations (and non-deviations such as redundant exclusion criteria) clearly and directly in the main text, in a separate paragraph or with a direct link to a separate document in the article.

5. *Explain yourself*. Disclosed deviations increase research transparency but a clarification for deviations facilitates the assessment of credibility, which was not the focus of the current study. Authors can also consider a sensitivity analysis that compares the planned analyses and the reported analyses, to assess the impact of the deviation.

6. *Use reporting guidelines*. Just like there are standards for various kinds of research articles, reporting guidelines for preregistered studies are necessary. With general guidelines, we agree upon which

deviations are important to disclosure, and which are trivial, and it should be easier for readers to understand which decisions were taken *a priori*, and which were not.

## 4.2. Limitations

First of all, there is a sense in which our assessment is not fair, because more vague preregistration plans and papers received the benefit of the doubt in case it was not clear whether the authors deviated from the plan. That the comparison between plan and paper is challenging, is also evident in the study by Heirene *et al.* [28]. For 40.6% of the scores, they could not determine adherence due to a lack of information in the preregistration plan, or both the plan and the paper. Our assessment was reviewed by the corresponding authors of the preregistered studies, but there are still disagreements left. Second, digging deeper into some materials, more specifically, in the data, might have resulted in different outcomes for some papers. For example, with raw data and analysis code, we would be able to see that all preregistered exclusion criteria are applied, even if they are not reported. However, we believe that such a level of scrutiny is uncalled for. Ideally, based on the paper alone, readers should be able to judge the extent to which researcher degrees of freedom influenced the results of the research. Third, we did not assess the impact of the deviations and treated all deviations as equal, while the consequences of deviations must differ. Finally, and perhaps most importantly, it should be kept in mind that our sample consists of a relatively small number of preregistered studies of the first generation from a single journal, and the generalizability of our findings is limited. The studies were conducted at a time when preregistration was rather new in psychology. Time is a very plausible game changer, and authors, reviewers, editors and the field of psychology in general are likely to improve how they deal with preregistration over time. The study by Heirene *et al.* supports this conjecture [28]. They found an increase in specificity and adherence over time in preregistered studies from 2017 to 2020. However, they argue that there is still room for improvement. Assessments like the current study are important to reflect on the present and future tools for preregistration.

## 5. Conclusion

Our work suggests that it is not obvious for preregistration to live up to its promise of making research results more transparent and more interpretable in the manuscript. This insight calls for better standards in preregistering and reporting preregistered research. It should, however, not be misinterpreted as a plea against the practice of preregistration. Neither does our work suggest that the results of the preregistered studies we assessed should not be trusted. We do not, and cannot, claim that the observed deviations have been deliberately unreported or constitute evidence of the exploitation of researcher degrees of freedom. The process of preregistering and reporting in accordance with the preregistration is tricky, and we realize that, given its relatively young status in psychology, the field collectively needs to go through a learning phase. It is thanks to the pioneering efforts of the authors of the papers in our sample and thanks to their courage to put themselves in a vulnerable position by preregistering their research and inviting scrutiny of their research, that we can learn about how we can do better as a field. Overall, we hope that our work will help speed up this learning phase. If researchers learn from the shortcomings currently observed in writing preregistration plans and reporting preregistered research, their research will benefit, and so will psychological science.

Ethics. No explicit institutional ethical approval was sought or provided for this study. While the authors include the details of correspondence with the corresponding authors of a number of studies, those correspondents gave written permission for their messages to be shared for this study.

Data accessibility. The assessments are stored on the OSF: https://osf.io/f49an/. The overview contains direct links to each assessed article and its corresponding preregistration plan(s). An R script to reproduce the main outcomes and figures 2 and 3 are available here: https://osf.io/bs5q2/.

Authors' contributions. Francis Tuerlinckx and Wolf Vanpaemel developed the study concept. Sara Gomes and Aline Claesen sampled and assessed the data under the supervision of Wolf Vanpaemel and Francis Tuerlinckx. Aline Claesen drafted the manuscript with revisions by Wolf Vanpaemel and Francis Tuerlinckx

Competing interests. We declare we have no competing interests.

Funding. The research leading to the results reported in this paper was supported in part by the Research Fund of KU Leuven (grant no. C14/19/054).

Acknowledgements. We wish to thank the corresponding authors of discussed preregistered studies for their comments. We also thank David Mellor, Stephen Lindsay, Brian Nosek, Daniel J. Simons, Cathrine Axfors, Rob Heirene and two anonymous reviewers for their valuable input.

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
