## [Peer Review File · Royal Society Open Science]

Review History

RSOS-211037.R0 (Original submission)

Review form: Reviewer 1

Is the manuscript scientifically sound in its present form?

Yes

Are the interpretations and conclusions justified by the results?

Yes

Is the language acceptable?

Yes

Do you have any ethical concerns with this paper?

No

Have you any concerns about statistical analyses in this paper?

No

Recommendation?

Accept as is

Comments to the Author(s)

In this paper, the authors study to what extent the first generation of preregistered studies in psychology (as measured from articles published with the preregistered badge in Psychological Science Feb 2015-Nov 2017) adheres to their preregistration plans and disclose deviations. They start with a sample of 23 articles and 38 preregistration plans, but because of lacking accessibility or lacking minimal methodological detail, the focus is on 16 articles and 27 corresponding articles. This decrease in sample size is very interesting in itself (and could maybe even be mentioned in the abstract?). The main outcome variable is the adherence of the study in the published article to the preregistration plan, as measured by six different items related to e.g. sample size and analysis that were coded as one of three options (no deviations, undisclosed deviation(s), or all deviations disclosed). The other two secondary outcome variables were template use and repository. The results suggest quite some non-adherence and lack of disclosure of deviations, in particular for sample size, exclusion criteria and analysis.

This is a very interesting and important paper. I am extremely positive and just have a few comments that the authors may or may not want to consider.

It would be interesting with some more discussion of how severe or problematic the deviations are. I understand that it is not the point of the study to explore whether e.g. non-preregistered control variables that are included lead to $p < 0.05$ results that otherwise are absent, but would this be possible/are the data for these studies available? If so perhaps that could be a separate study?

For the "Statistical analysis" section on p.7 starting on line 30, it would be interesting with more information on deviations.

When it comes to the recommendations, I am curious about what the authors think about changing the norm of how we write papers to make them more like Registered Reports (even when they are not RRs)? This format would make it easier to show and disclose deviations from preregistration plans etc.

I really like the Conclusion section and how the authors acknowledge the efforts of the original authors of the studies they sampled.

There are other papers comparing preregistrations with published papers, e.g. Ofosu and Posner (2021). I think it would make sense to include a discussion of these papers.

p.9, line 42: a "d" is missing in "and" in "Finally, an...".

George K. Ofosu, Daniel N. Posner (2021). Pre-Analysis Plans: An Early Stocktaking. Accepted, Perspectives on Politics.

Review form: Reviewer 2 (Cathrine Axfors)

Is the manuscript scientifically sound in its present form?

Yes

Are the interpretations and conclusions justified by the results?

Yes

Is the language acceptable?

Yes

Do you have any ethical concerns with this paper?

No

Have you any concerns about statistical analyses in this paper?

No

Recommendation?

Accept with minor revision (please list in comments)

Comments to the Author(s)

Dear Authors and Editor,

Thank you for the chance to review this interesting piece of work. The article describes, in a sample of studies that received the Preregistered badge in Psychological Science (2015-2017), the extent of (disclosed and undisclosed) deviations from preregistration plans in the final publication. Corresponding authors of all studies were contacted to obtain comments on the assessments. The authors find that the vast majority of publications included undisclosed deviations, and identify main areas where deviations were more often observed.

This is one of very few articles to assess differences between preregistrations and publication, and I deem it likely very interesting to the readers. Main shortcomings are in my view the intrinsic difficulties of the assessments, and that level of specificity of the preregistered plans is not addressed. Please find below additional comments, which I hope will be of help.

Sincerely,
Cathrine Axfors
MD, PhD

--

General comments:

1. I agree with the limitation stated that underspecified plans might escape scrutiny and this would be a lamentable development. I would encourage the authors to consider adding an assessment of the level of detail of the preregistered plan. In another study, <https://psyarxiv.com/nj4es>, Heirene et al believed this was relevant when they assessed 53 preregistered plans and so "scored their level of specificity using a 23-item protocol developed to measure the extent to which a clear and exhaustive preregistration plan restricts various researcher degrees of freedom (RDoF; i.e., the many methodological choices available to researchers when collecting and analysing data, and when reporting their findings)."
2. The Comments from corresponding authors part is almost handled as a quality control step, while I would encourage the authors to emphasize this step as another data collection. Especially the part about reasons for deviations could perhaps be described a bit more systematically/comprehensively (e.g., categorize the reasons for deviations in a table, give examples, and perhaps to count the number of plans/deviations for which that reason applies).
3. I believe that what is currently described as exclusion criteria are added measurements, which make the paper more interesting. Page 4, Row 15: That many articles were found not accessible or not having enough detail to be assessed for adherence are findings in themselves. I recommend the authors to reformulate so that the study is described as having three parts. In the first (accessibility), there are no exclusion criteria. In the second (minimal detail), there are. In the

third (adherence), yet additional ones are added. The third part contains the main outcome. The current grouping into exclusion criteria and main analysis is a bit confusing, especially since the full initial sample (38 preregistration plans) is referred in the secondary outcomes. Also, with this, I suggest that a flowchart be added to describe the exclusions at different steps. The heading "Exclusions" would be changed to "Accessibility and detail" or a better suited one.

4. It would have been of top interest to ascertain whether studies were indeed preregistered. It is salient (although perhaps unsurprising) that the authors deemed this infeasible. For clinical trials, it is often stated in registrations when the first patient was enrolled. I encourage the authors to add a sentence or two in the discussion about this and any thoughts on how such ascertainment could be implemented for observational studies (at other some stage of the publication process, if not now).

Specific comments:

5. Could the circumstances around badges be explained briefly? Are all the studies in the sample performing confirmatory hypothesis testing? I assume so, but it would be relevant to state. How were changes from plans to publication categorized that were explicitly described as exploratory (if such were observed)?

6. Page 4, Row 57 onwards. All exclusion criteria and "primary measures" were assessed by two independent raters. What are the primary measures referred here? If referring to main outcome, the last sentence on the same page indicates the opposite: "Adherence was assessed by one rater", i.e., not independently by two raters. What does it mean that adherence was assessed "under the supervision" of two co-authors? Asking since adherence is the main outcome, and as the authors write, it proved to be a far from trivial task.

7. Page 5, row 34. "Did not include a planned sample size". Were there any studies without a planned sample size, with a rationale for why the sample size was not prespecified?

8. Page 5, row 38. Here, like in some other parts of the manuscript, authors add their observations and experiences as they go. While I can recognize the need to disclose such details from the perspective of being an author, as a reader I do not always appreciate the side details before having understood the main picture. I would encourage the authors to refrain from interpreting the findings before the discussion section, and to place for example the content under "preliminary observations" in the limitations section of the discussion.

9. Page 5, last row, "there were no deviations fully disclosed in the article for nine out of 25..." Suggest to reformulate to "None of the deviations were fully disclosed in the article for nine out of 25 preregistered studies with deviations", for readability. And, "In the remaining 16 ... studies, at least one [of the] deviations was fully disclosed in the article."

10. Figure 1. Layout and editing. The study names are a bit confusing for readers who don't revisit the assessment data file, but this is a minor point.

11. To add to methods section: How are "variables" defined and what constitutes deviations from "variables"? Since the adherence criteria didn't specify operationalization of variables. Is it the number of variables, or their description? Generally, the description on page 4, row 30, of adherence items lacks details.

12. Probably related: What did the authors mean with that a variable is operationalized? "Operationalization of the variables" was required for inclusion in the adherence analysis, but that item seems to be baked into the "variables" item in the very adherence analysis.

13. It would add strength to the manuscript if the authors discussed their findings in relation to similar studies. From Wikipedia, heading Preregistration (science), there seem to exist at least one or two, "... researchers rarely follow the exact research methods and analyses that they preregister (Abrams et al., 2020; Claesen et al., 2019; Heirene et al., 2021; see also Boghdadly et al., 2018; Singh et al., 2021; Sun et al., 2019)". Some of these are for RCTs, but comparisons can still be relevant. Also, a comment on how the methods here differed from the Heirene et al study (and limitations/strengths in light of that) would be excellent in my opinion.

14. About generalizability. Could the authors add in the discussion how many articles have received the preregistered badge during the years after 2017? Just to give the reader a sense of the development and how big a portion is assessed here.

15. It is a challenge to strike the right balance of specificity and sensitivity in the definition of deviations. For sample size, the authors comment that it may be trivial with a small deviation from the planned number, a view endorsed by some corresponding authors which I sympathize with. For details that were "compatible" with the plan but represented an added level of granularity, perhaps these escaped the check too lightly (see comment 1). Back to sample size: just because this is a number (and so a difference is easier to detect), it's not sure that it's a relevant difference, much like the case with statistically significant results in a very large dataset. Do the authors believe that a priori sample size specification is relevant for all the articles assessed? If not, perhaps that item should be reconsidered to allow some percentage change as no deviation.

16. Personal preference: parts of the results that describe methodology (how sample size, exclusion criteria, and statistical analysis were operationalized) I would prefer to see in the methods section. At present, the description of the results mixes methods details that have previously not been disclosed, and interpretations, with the results reporting. I admit that differences may exist between research fields, and I come from a tradition with more separated sections where concise is key.

17. The authors have done great work. I was surprised by the at times defensive tone, and that the authors seem to have felt the need to excuse their definitions or entire endeavour. Then I read the comments from some of the contacted corresponding authors, which are in my view quite unforgiving. Most readers will probably agree that this type of analysis is a necessary next step after introducing preregistration as a phenomenon, and I don't believe it is the responsibility of the authors to bolster all reactions.

Review form: Reviewer 3 (Robert Heirene)

Is the manuscript scientifically sound in its present form?

Yes

Are the interpretations and conclusions justified by the results?

Yes

Is the language acceptable?

Yes

Do you have any ethical concerns with this paper?

No

Have you any concerns about statistical analyses in this paper?

No

Recommendation?

Major revision is needed (please make suggestions in comments)

Comments to the Author(s)

Thursday, 5 August 2021

OVERALL COMMENTS

Thank you for the opportunity to review this interesting manuscript. The findings presented advance the field of psychology and beyond, the conclusions are supported by the data, and the methods appear scientifically sound. There are several concerns that I have (e.g., the failure to contextualise this study and the outcomes with the existing literature) but the authors should be able to address these in a revision. Below I provide specific comments on each of the individual sections in the manuscript.

INTRODUCTION**FIRST PARAGRAPH**

The authors state: “During the research process, researchers inevitably make numerous decisions, collectively known as researcher degrees of freedom” – please provide examples of such decisions for researchers less familiar with the concept of researcher degrees of freedom

The authors state: “One way to increase research transparency is to preregister research, which involves freezing the analytic choices on a public third-party repository prior to seeing, or ideally prior to collecting, the data [9]. By specifying decisions before data collection, researcher degrees of freedom are restricted, and decisions that are made during the data collection and analysis cannot be mistakenly reported as a priori.” – for the purpose of clarity it would be worth noting that researchers often preregister more than just their analytical decisions (e.g., hypotheses, study design etc.), even if these elements are not the focus of this paper.

SECOND PARAGRAPH

There is a study that has looked at adherence to preregistrations in the political science and economics literature by Osfu & Posner (2019) – Ofosu, G., & Posner, D. N. (2019). Pre-analysis Plans: A Stocktaking (Pre-Print). MetaArXiv. <https://doi.org/10.31222/osf.io/e4pum>

It would be helpful if the authors could briefly discuss this study here to better contextualise their study and describe the current state of existing research on this topic (or lack thereof)

Similarly, my colleagues and I have just pre-printed a study on this topic which is highly relevant (although this was very recent and therefore the authors will not have had time to incorporate this into their article). – Heirene, R., LaPlante, D., Louderback, E. R., Keen, B., Bakker, M., Serafimovska, A., & Gainsbury, S. M. (2021, July 16). Preregistration specificity & adherence: A review of preregistered gambling studies & cross-disciplinary comparison. <https://doi.org/10.31234/osf.io/nj4es>

FINAL PARAGRAPH

It would be helpful if the authors more clearly stated the aim of the study towards the end of this paragraph (if space is an issue, then the sentences describing the findings/ conclusions can be removed, but this is more of a personal preference than a genuine issue).

METHODS

There is no information on data analysis here – I recognise that the authors only include summary statistics and don't report the results of any inferential statistical tests, but it would still

be useful for the authors to state what software+packages they used to analyse the data and develop figures and where we can access the analysis code (if available?)

DISCLOSURE SUBSECTION

While the reasons stated by the authors provide a reasonable rationale for why the study was not preregistered, it may appear odd to those who are less familiar with the process of preregistration (or who are critical of it) that the authors did not preregister their own study on preregistration practices. Perhaps this could be avoided by discussing the purpose and use of preregistration to a greater extent in the introductory section (e.g., use in confirmatory/hypothesis testing research vs. exploratory studies).

Can the authors clarify what is meant by “open practices disclosure items” please?

I can't access the link to the assessment procedure. I have requested access to this from the authors and would like to review it before finalising my review. Is there any reason why this assessment is not available to the public? Especially given that the study has been available for some time as a preprint?

SAMPLE SUBSECTION

Can the authors better describe the characteristics of the preregistrations and studies included, please? For example, how many studies were published in each of the three years studied (2015, 2016, 2017)? What types of studies were included and did the authors exclude systematic reviews and RCTs (which have different registration requirements)? Where were the studies preregistered (OSF, PROSPERO, aspredicted)? What preregistration templates were used? Both research by myself and colleagues and Bakker et al. (2021; doi:10.1371/journal.pbio.3000937) have shown the templates used affect the quality of the preregistration? I recognise that the authors were not looking at the quality of the preregistration itself, but it could still be interesting from an adherence perspective. Perhaps a table could be used to display most of this information concisely? (ADDITION: I see the authors describe some of these variables in the secondary outcomes; still, perhaps this information could be more easily summarised as a table?)

MEASURES AND OUTCOMES SUBSECTION

It's great that the authors have described their process of screening papers/preregistrations very clearly at the beginning of this section.

Minor issue: the authors jump from past tense to present tense (this sentence “Without full disclosure, deviations are undisclosed.”) then back to past tense again in the second paragraph of this section.

PROCEDURE SUBSECTION

Having just performed a very similar study, I find it concerning that only one researcher scored adherence to preregistrations. In our study, two researchers independently scored preregistration adherence and found it extremely difficult and time-consuming to make these comparisons – we spent almost as much time reviewing and discussing our scoring as we did actually doing the scoring due to ambiguities and inconsistencies between preregistrations and papers in terminology, phrasing, structure etc. and clearly the rater faced similar difficulties based on the statements in the second paragraph of this section. Can the authors provide a bit more information about they avoided errors in this process and how the sole rater was supervised (e.g., were any of the scores directly checked by another author)?

The authors state: “As a result, there were cases in which it was difficult to assess whether there was a deviation from the preregistration plan. Whenever there was reasonable doubt about a deviation from the preregistration plan, this was coded as no deviations” – were the number of instances when this occurred and recorded? I think if the team could not adequately tell whether authors of their sample papers had deviated or not from their registration then this is a separate and noteworthy phenomenon. As the authors discuss later, their scoring process also rewards poor specification by not separating out these cases. Not differentiating these instances concerns me as when scoring adherence on 20+ items in our paper, nearly 41% of the scores we gave were

classified “Unable to tell [due to a lack of detail in the pre-registration, paper, or both]” (<https://psyarxiv.com/nj4es/>). Admittedly, this figure is likely to be lower in the present study as the authors required preregistrations to contain a minimal set of details (we did not) that were then compared.

RESULTS

PRELIMINARY OBSERVATIONS SUBSECTION

This section is an important addition and mirrors many of the issues we faced when making these comparisons. This qualitative/process related detail can be lost in this sort of study and so I was very pleased to see the authors include this here in the findings.

PRIMARY OUTCOME: ADHERENCE SUBSECTION

I finally got access to the assessment procedures and raw data from the link included in Figure 1’s note. I downloaded and reanalysed the data to reproduce the outcomes reported in this subsection. I successfully reproduced all outcomes.

I have included the code used for this below (appendix) and I attached the full script & html output as documents to this review.

I find the X axis labels on Figure 1 to be very confusing. Can the authors try and make this more readable without having to look at the raw data (although even then I find it confusing)? Just changing the labels to “paper 21 (2 studies)” would be easier to understand.

SAMPLE SIZE SUBSECTION

It seems like there is a lot of information reported here for little gain. The authors could much more concisely say that they treated any discrepancy between the exact sample size preregistered and the sample size reported in the paper as a deviation, without the need to provide a lengthy example to illustrate this.

It would be useful if the office can provide some more qualitative summary information here rather than detailed qualitative examples. For instance, what was the average/median/range for the discrepancy between preregistered and actual sample size? Was there any pattern in the studies that disclosed the deviation (e.g., smaller discrepancies?). Overall, it seemed more like I was reading a discussion+recommendations section here rather than a description of the results. Statistical analysis subsection

Same issues as stated above apply here. For example, the following statement appears odd in a results section and should appear in the discussion: “Detailing the statistical analysis is probably the hardest part of writing a preregistration plan. It is, therefore, no surprise that especially for the statistical analysis, deviations are likely to occur.”

Why don’t the authors provide a subsection for the other 3 areas of adherence studied (i.e., variables, hypotheses, and procedures)? I recognise these areas had the most overall deviations, but there were as many undeclared deviations relating to hypotheses as there were relating to sample sizes

EXCLUSION CRITERIA SUBSECTION

Again, there is a very lengthy quotation from one of the papers included here that could be cut down substantially. Again, there is a similar feel to this paragraph like it is more of a discussion section or form a conceptual paper, rather than being a description of the results.

COMMENTS FROM CORRESPONDING AUTHORS

This is another interesting addition to this paper, and I commend the authors for contacting all authors in their sample.

DISCUSSION

A comparison of the findings presented here with those from the existing literature in this area would be beneficial here before moving on to offer recommendations, especially as later in this authors state: "Time is a very plausible game changer, and it is likely that authors, reviewers, and editors are improving how they deal with preregistration over time. New assessments might reveal an improvement compared to what we found in this sample. However, this assessment is important to reflect on current and new tools for preregistration." – more recent assessments are now available.

The authors state: "Especially the sample size and analysis plan were difficult to assess due to a lack of standards." – can you clarify what is meant by "and lack of standards" here please?

The recommendations provided are a great addition to this paper and will be of significant benefit to the field if abided by. The 6th recommendation seems like it could be more instructive – what guidelines would the authors recommend based on their findings?

The sample size used was relatively small (admittedly beyond their controls) and from papers published in a single journal. The authors should comment on the limited external validity of their findings in the limitations section.

ABSTRACT

It will be beneficial for the author to include some of the key outcomes in the abstract (e.g., the exact number of studies with un/disclosed deviations)

I hope the authors find the above comments useful in revising their manuscript for publication.

Sincerely,
Rob Heirene

Appendix

R script used to reanalyse data:

```
---
title: Re-analysis for review
author: "Rob Heirene"
date: "`r format(Sys.time(), '%d %B, %Y')`"
output: html_document
---
# -----

```{r message=FALSE, results = FALSE}
library("dplyr")
```

Remove all objects from work space before starting:
```{r}
rm(list = ls())
unlink("Workspace_prereg_study_final_analyses_FINAL.RData")
```

```{r}
Load data:
data<- read.csv("Summary_responses_final.csv") # I extracted just the tab titled "adherence" for
this

View dataset:
```

```
View(data)
names(data)
```

```
Make variable names easier to work with:
data1 <- data %>% rename(none = "Number.of.aspects.with.no.deviations",
 all_disclosed = "Number.of.aspects.with.all.deviations.disclosed",
 undisclosed = "Number.aspects.with.undisclosed.deviations")
```

```
...
```

Now I've loaded the data I will check outcomes reported in first paragraph of primary outcome subsection of results:

The authors state here:

- "In our sample, two of the 27 (7%) selected studies did not deviate from the preregistered plan in any of the preregistered methodological aspects (see Figure 1). One study reported all the deviations. In the remaining 24 of 27 (89%) studies, there was at least one item for which a discrepancy between the preregistration plan and the journal article was not fully disclosed. There were no deviations fully disclosed in the article for nine out of 25 (36%) preregistered studies with deviations. In the remaining 16 (64%) studies, at least one deviation was fully disclosed in the article."

```
```{r}
```

```
data1 %>% filter(none == "6") # 6 = no deviations at all
# Percentage presented = 7%:
2/27*100
```

```
data1 %>% filter(undisclosed > 0) %>% nrow() # anything > 1 indicates an UNdeclared
disclosure
# Percentage presented = 89%:
24/27*100
```

```
data1 %>% filter(all_disclosed > 0) %>% nrow() # anything > 1 indicates a declared disclosure
```

```
# Percentage presented = 64% (they calculate out of the 25 with deviations when reporting this):
16/25*100
```

```
```
```

Now the second paragraph of this section where the authors state:

- "Non-adherence was most prevalent for the planned sample size, exclusion criteria, and analysis. For six out of 27 (22%) studies, the deviation in sample size is disclosed, as opposed to five (19%) studies, where the deviation is left undisclosed. For five (19%) studies, all changes in exclusion criteria are reported, while in 12 (44%) other studies this is not the case. Finally, in five (19%) studies all deviations in the statistical analysis are reported, but in 13 (48%) not. For this reason, we will discuss some common issues observed for these three aspects."

```
```{r}
```

```
# Hypothesis/research question:
table(data$H.RQ)
```

```
table(data$Variables)
```

```
table(data$Sample.size) # seems right. Percentages:
```

```
6/27*100
```

```
5/27*100
```

```
table(data$Exclusion.criteria) # seems right. Percentages:
```

```
5/27*100
```

```
12/27*100
```

```
table(data$Procedure)
```

```
table(data$Analysis) # seems right. Percentages:
```

```
5/27*100
```

```
13/27*100
```

```
```
```

## Decision letter (RSOS-211037.R0)

Dear Ms Claesen

On behalf of the Editors, we are pleased to inform you that your Manuscript RSOS-211037 "Comparing Dream to Reality: An Assessment of Adherence of the First Generation of Preregistered Studies" has been accepted for publication in Royal Society Open Science subject to minor revision in accordance with the referees' reports. Please find the referees' comments along with any feedback from the Editors below my signature.

Please submit your revised manuscript and required files (see below) no later than 7 days from today's (ie 17-Aug-2021) date. Note: the ScholarOne system will 'lock' if submission of the revision is attempted 7 or more days after the deadline. If you do not think you will be able to meet this deadline please contact the editorial office immediately.

on behalf of Professor Zoltan Dienes (Associate Editor) and Essi Viding (Subject Editor)  
openscience@royalsociety.org

Associate Editor Comments to Author (Professor Zoltan Dienes):

Associate Editor: 1

Comments to the Author:

I have three excellent and very thoughtful reviews back; all are highly positive about your paper but also providing important points to consider in preparing a revision. I look forward to seeing the revised manuscript.

Reviewer comments to Author:

Reviewer: 1

Comments to the Author(s)

In this paper, the authors study to what extent the first generation of preregistered studies in psychology (as measured from articles published with the preregistered badge in Psychological Science Feb 2015-Nov 2017) adheres to their preregistration plans and disclose deviations. They start with a sample of 23 articles and 38 preregistration plans, but because of lacking accessibility or lacking minimal methodological detail, the focus is on 16 articles and 27 corresponding articles. This decrease in sample size is very interesting in itself (and could maybe even be mentioned in the abstract?). The main outcome variable is the adherence of the study in the published article to the preregistration plan, as measured by six different items related to e.g. sample size and analysis that were coded as one of three options (no deviations, undisclosed deviation(s), or all deviations disclosed). The other two secondary outcome variables were template use and repository. The results suggest quite some non-adherence and lack of disclosure of deviations, in particular for sample size, exclusion criteria and analysis.

This is a very interesting and important paper. I am extremely positive and just have a few comments that the authors may or may not want to consider.

It would be interesting with some more discussion of how severe or problematic the deviations are. I understand that it is not the point of the study to explore whether e.g. non-preregistered control variables that are included lead to  $p < 0.05$  results that otherwise are absent, but would this be possible/are the data for these studies available? If so perhaps that could be a separate study?

For the "Statistical analysis" section on p.7 starting on line 30, it would be interesting with more information on deviations.

When it comes to the recommendations, I am curious about what the authors think about changing the norm of how we write papers to make them more like Registered Reports (even when they are not RRs)? This format would make it easier to show and disclose deviations from preregistration plans etc.

I really like the Conclusion section and how the authors acknowledge the efforts of the original authors of the studies they sampled.

There are other papers comparing preregistrations with published papers, e.g. Ofosu and Posner (2021). I think it would make sense to include a discussion of these papers.

p.9, line 42: a “d” is missing in “and” in “Finally, an...”.

George K. Ofosu, Daniel N. Posner (2021). Pre-Analysis Plans: An Early Stocktaking. Accepted, Perspectives on Politics.

Reviewer: 2

Comments to the Author(s)

Dear Authors and Editor,

Thank you for the chance to review this interesting piece of work. The article describes, in a sample of studies that received the Preregistered badge in Psychological Science (2015-2017), the extent of (disclosed and undisclosed) deviations from preregistration plans in the final publication. Corresponding authors of all studies were contacted to obtain comments on the assessments. The authors find that the vast majority of publications included undisclosed deviations, and identify main areas where deviations were more often observed.

This is one of very few articles to assess differences between preregistrations and publication, and I deem it likely very interesting to the readers. Main shortcomings are in my view the intrinsic difficulties of the assessments, and that level of specificity of the preregistered plans is not addressed. Please find below additional comments, which I hope will be of help.

Sincerely,  
Cathrine Axfors  
MD, PhD

--

General comments:

1. I agree with the limitation stated that underspecified plans might escape scrutiny and this would be a lamentable development. I would encourage the authors to consider adding an assessment of the level of detail of the preregistered plan. In another study, <https://psyarxiv.com/nj4es>, Heirene et al believed this was relevant when they assessed 53 preregistered plans and so “scored their level of specificity using a 23-item protocol developed to measure the extent to which a clear and exhaustive preregistration plan restricts various researcher degrees of freedom (RDoF; i.e., the many methodological choices available to researchers when collecting and analysing data, and when reporting their findings).”

2. The Comments from corresponding authors part is almost handled as a quality control step, while I would encourage the authors to emphasize this step as another data collection. Especially the part about reasons for deviations could perhaps be described a bit more systematically/comprehensively (e.g., categorize the reasons for deviations in a table, give examples, and perhaps to count the number of plans/deviations for which that reason applies).

3. I believe that what is currently described as exclusion criteria are added measurements, which make the paper more interesting. Page 4, Row 15: That many articles were found not accessible or not having enough detail to be assessed for adherence are findings in themselves. I recommend the authors to reformulate so that the study is described as having three parts. In the first (accessibility), there are no exclusion criteria. In the second (minimal detail), there are. In the third

(adherence), yet additional ones are added. The third part contains the main outcome. The current grouping into exclusion criteria and main analysis is a bit confusing, especially since the full initial sample (38 preregistration plans) is referred in the secondary outcomes. Also, with this, I suggest that a flowchart be added to describe the exclusions at different steps. The heading "Exclusions" would be changed to "Accessibility and detail" or a better suited one.

4. It would have been of top interest to ascertain whether studies were indeed preregistered. It is salient (although perhaps unsurprising) that the authors deemed this infeasible. For clinical trials, it is often stated in registrations when the first patient was enrolled. I encourage the authors to add a sentence or two in the discussion about this and any thoughts on how such ascertainment could be implemented for observational studies (at other some stage of the publication process, if not now).

Specific comments:

5. Could the circumstances around badges be explained briefly? Are all the studies in the sample performing confirmatory hypothesis testing? I assume so, but it would be relevant to state. How were changes from plans to publication categorized that were explicitly described as exploratory (if such were observed)?

6. Page 4, Row 57 onwards. All exclusion criteria and "primary measures" were assessed by two independent raters. What are the primary measures referred here? If referring to main outcome, the last sentence on the same page indicates the opposite: "Adherence was assessed by one rater", i.e., not independently by two raters. What does it mean that adherence was assessed "under the supervision" of two co-authors? Asking since adherence is the main outcome, and as the authors write, it proved to be a far from trivial task.

7. Page 5, row 34. "Did not include a planned sample size". Were there any studies without a planned sample size, with a rationale for why the sample size was not prespecified?

8. Page 5, row 38. Here, like in some other parts of the manuscript, authors add their observations and experiences as they go. While I can recognize the need to disclose such details from the perspective of being an author, as a reader I do not always appreciate the side details before having understood the main picture. I would encourage the authors to refrain from interpreting the findings before the discussion section, and to place for example the content under "preliminary observations" in the limitations section of the discussion.

9. Page 5, last row, "there were no deviations fully disclosed in the article for nine out of 25..." Suggest to reformulate to "None of the deviations were fully disclosed in the article for nine out of 25 preregistered studies with deviations", for readability. And, "In the remaining 16 ... studies, at least one [of the] deviations was fully disclosed in the article."

10. Figure 1. Layout and editing. The study names are a bit confusing for readers who don't revisit the assessment data file, but this is a minor point.

11. To add to methods section: How are "variables" defined and what constitutes deviations from "variables"? Since the adherence criteria didn't specify operationalization of variables. Is it the number of variables, or their description? Generally, the description on page 4, row 30, of adherence items lacks details.

12. Probably related: What did the authors mean with that a variable is operationalized? "Operationalization of the variables" was required for inclusion in the adherence analysis, but that item seems to be baked into the "variables" item in the very adherence analysis.

13. It would add strength to the manuscript if the authors discussed their findings in relation to similar studies. From Wikipedia, heading Preregistration (science), there seem to exist at least one or two, "... researchers rarely follow the exact research methods and analyses that they preregister (Abrams et al., 2020; Claesen et al., 2019; Heirene et al., 2021; see also Boghdadly et al., 2018; Singh et al., 2021; Sun et al., 2019)". Some of these are for RCTs, but comparisons can still be relevant. Also, a comment on how the methods here differed from the Heirene et al study (and limitations/strengths in light of that) would be excellent in my opinion.

14. About generalizability. Could the authors add in the discussion how many articles have received the preregistered badge during the years after 2017? Just to give the reader a sense of the development and how big a portion is assessed here.

15. It is a challenge to strike the right balance of specificity and sensitivity in the definition of deviations. For sample size, the authors comment that it may be trivial with a small deviation from the planned number, a view endorsed by some corresponding authors which I sympathize with. For details that were "compatible" with the plan but represented an added level of granularity, perhaps these escaped the check too lightly (see comment 1). Back to sample size: just because this is a number (and so a difference is easier to detect), it's not sure that it's a relevant difference, much like the case with statistically significant results in a very large dataset. Do the authors believe that a priori sample size specification is relevant for all the articles assessed? If not, perhaps that item should be reconsidered to allow some percentage change as no deviation.

16. Personal preference: parts of the results that describe methodology (how sample size, exclusion criteria, and statistical analysis were operationalized) I would prefer to see in the methods section. At present, the description of the results mixes methods details that have previously not been disclosed, and interpretations, with the results reporting. I admit that differences may exist between research fields, and I come from a tradition with more separated sections where concise is key.

17. The authors have done great work. I was surprised by the at times defensive tone, and that the authors seem to have felt the need to excuse their definitions or entire endeavour. Then I read the comments from some of the contacted corresponding authors, which are in my view quite unforgiving. Most readers will probably agree that this type of analysis is a necessary next step after introducing preregistration as a phenomenon, and I don't believe it is the responsibility of the authors to bolster all reactions.

Reviewer: 3  
Comments to the Author(s)  
Royal Society Open Science  
Thursday, 5 August 2021

#### OVERALL COMMENTS

Thank you for the opportunity to review this interesting manuscript. The findings presented advance the field of psychology and beyond, the conclusions are supported by the data, and the methods appear scientifically sound. There are several concerns that I have (e.g., the failure to contextualise this study and the outcomes with the existing literature) but the authors should be able to address these in a revision. Below I provide specific comments on each of the individual sections in the manuscript.

#### INTRODUCTION FIRST PARAGRAPH

The authors state: “During the research process, researchers inevitably make numerous decisions, collectively known as researcher degrees of freedom” – please provide examples of such decisions for researchers less familiar with the concept of researcher degrees of freedom

The authors state: “One way to increase research transparency is to preregister research, which involves freezing the analytic choices on a public third-party repository prior to seeing, or ideally prior to collecting, the data [9]. By specifying decisions before data collection, researcher degrees of freedom are restricted, and decisions that are made during the data collection and analysis cannot be mistakenly reported as a priori.” – for the purpose of clarity it would be worth noting that researchers often preregister more than just their analytical decisions (e.g., hypotheses, study design etc.), even if these elements are not the focus of this paper.

#### SECOND PARAGRAPH

There is a study that has looked at adherence to preregistrations in the political science and economics literature by Osfu & Posner (2019) – Ofosu, G., & Posner, D. N. (2019). Pre-analysis Plans: A Stocktaking (Pre-Print). MetaArXiv. <https://doi.org/10.31222/osf.io/e4pum>

It would be helpful if the authors could briefly discuss this study here to better contextualise their study and describe the current state of existing research on this topic (or lack thereof)

Similarly, my colleagues and I have just pre-printed a study on this topic which is highly relevant (although this was very recent and therefore the authors will not have had time to incorporate this into their article). – Heirene, R., LaPlante, D., Louderback, E. R., Keen, B., Bakker, M., Serafimovska, A., & Gainsbury, S. M. (2021, July 16). Preregistration specificity & adherence: A review of preregistered gambling studies & cross-disciplinary comparison. <https://doi.org/10.31234/osf.io/nj4es>

#### FINAL PARAGRAPH

It would be helpful if the authors more clearly stated the aim of the study towards the end of this paragraph (if space is an issue, then the sentences describing the findings/ conclusions can be removed, but this is more of a personal preference than a genuine issue).

#### METHODS

There is no information on data analysis here – I recognise that the authors only include summary statistics and don’t report the results of any inferential statistical tests, but it would still be useful for the authors to state what software+packages they used to analyse the data and develop figures and where we can access the analysis code (if available?)

#### DISCLOSURE SUBSECTION

While the reasons stated by the authors provide a reasonable rationale for why the study was not preregistered, it may appear odd to those who are less familiar with the process of preregistration (or who are critical of it) that the authors did not preregister their own study on preregistration practices. Perhaps this could be avoided by discussing the purpose and use of preregistration to a greater extent in the introductory section (e.g., use in confirmatory/hypothesis testing research vs. exploratory studies).

Can the authors clarify what is meant by “open practices disclosure items” please?

I can’t access the link to the assessment procedure. I have requested access to this from the authors and would like to review it before finalising my review. Is there any reason why this assessment is not available to the public? Especially given that the study has been available for some time as a preprint?

#### SAMPLE SUBSECTION

Can the authors better describe the characteristics of the preregistrations and studies included, please? For example, how many studies were published in each of the three years studied (2015, 2016, 2017)? What types of studies were included and did the authors exclude systematic reviews and RCTs (which have different registration requirements)? Where were the studies preregistered (OSF, PROSPERO, aspredicted)? What preregistration templates were used? Both research by

myself and colleagues and Bakker et al. (2021; doi:10.1371/journal.pbio.3000937) have shown the templates used affect the quality of the preregistration? I recognise that the authors were not looking at the quality of the preregistration itself, but it could still be interesting from an adherence perspective. Perhaps a table could be used to display most of this information concisely? (ADDITION: I see the authors describe some of these variables in the secondary outcomes; still, perhaps this information could be more easily summarised as a table?)

#### MEASURES AND OUTCOMES SUBSECTION

It's great that the authors have described their process of screening papers/preregistrations very clearly at the beginning of this section.

Minor issue: the authors jump from past tense to present tense (this sentence "Without full disclosure, deviations are undisclosed.") then back to past tense again in the second paragraph of this section.

#### PROCEDURE SUBSECTION

Having just performed a very similar study, I find it concerning that only one researcher scored adherence to preregistrations. In our study, two researchers independently scored preregistration adherence and found it extremely difficult and time-consuming to make these comparisons – we spent almost as much time reviewing and discussing our scoring as we did actually doing the scoring due to ambiguities and inconsistencies between preregistrations and papers in terminology, phrasing, structure etc. and clearly the rater faced similar difficulties based on the statements in the second paragraph of this section. Can the authors provide a bit more information about they avoided errors in this process and how the sole rater was supervised (e.g., were any of the scores directly checked by another author)?

The authors state: "As a result, there were cases in which it was difficult to assess whether there was a deviation from the preregistration plan. Whenever there was reasonable doubt about a deviation from the preregistration plan, this was coded as no deviations " – were the number of instances when this occurred and recorded? I think if the team could not adequately tell whether authors of their sample papers had deviated or not from their registration then this is a separate and noteworthy phenomenon. As the authors discuss later, their scoring process also rewards poor specification by not separating out these cases. Not differentiating these instances concerns me as when scoring adherence on 20+ items in our paper, nearly 41% of the scores we gave were classified "Unable to tell [due to a lack of detail in the pre-registration, paper, or both]" (<https://psyarxiv.com/nj4es/>). Admittedly, this figure is likely to be lower in the present study as the authors required preregistrations to contain a minimal set of details (we did not) that were then compared.

### RESULTS

#### PRELIMINARY OBSERVATIONS SUBSECTION

This section is an important addition and mirrors many of the issues we faced when making these comparisons. This qualitative/process related detail can be lost in this sort of study and so I was very pleased to see the authors include this here in the findings.

#### PRIMARY OUTCOME: ADHERENCE SUBSECTION

I finally got access to the assessment procedures and raw data from the link included in Figure 1's note. I downloaded and reanalysed the data to reproduce the outcomes reported in this subsection. I successfully reproduced all outcomes.

I have included the code used for this below (appendix) and I attached the full script & html output as documents to this review.

I find the X axis labels on Figure 1 to be very confusing. Can the authors try and make this more readable without having to look at the raw data (although even then I find it confusing)? Just changing the labels to "paper 21 (2 studies)" would be easier to understand.

### SAMPLE SIZE SUBSECTION

It seems like there is a lot of information reported here for little gain. The authors could much more concisely say that they treated any discrepancy between the exact sample size preregistered and the sample size reported in the paper as a deviation, without the need to provide a lengthy example to illustrate this.

It would be useful if the office can provide some more qualitative summary information here rather than detailed qualitative examples. For instance, what was the average/median/range for the discrepancy between preregistered and actual sample size? Was there any pattern in the studies that disclosed the deviation (e.g., smaller discrepancies?). Overall, it seemed more like I was reading a discussion+recommendations section here rather than a description of the results.

### Statistical analysis subsection

Same issues as stated above apply here. For example, the following statement appears odd in a results section and should appear in the discussion: "Detailing the statistical analysis is probably the hardest part of writing a preregistration plan. It is, therefore, no surprise that especially for the statistical analysis, deviations are likely to occur."

Why don't the authors provide a subsection for the other 3 areas of adherence studied (i.e., variables, hypotheses, and procedures)? I recognise these areas had the most overall deviations, but there were as many undeclared deviations relating to hypotheses as there were relating to sample sizes

### EXCLUSION CRITERIA SUBSECTION

Again, there is a very lengthy quotation from one of the papers included here that could be cut down substantially. Again, there is a similar feel to this paragraph like it is more of a discussion section or form a conceptual paper, rather than being a description of the results.

### COMMENTS FROM CORRESPONDING AUTHORS

This is another interesting addition to this paper, and I commend the authors for contacting all authors in their sample.

### DISCUSSION

A comparison of the findings presented here with those from the existing literature in this area would be beneficial here before moving on to offer recommendations, especially as later in this authors state: "Time is a very plausible game changer, and it is likely that authors, reviewers, and editors are improving how they deal with preregistration over time. New assessments might reveal an improvement compared to what we found in this sample. However, this assessment is important to reflect on current and new tools for preregistration." – more recent assessments are now available.

The authors state: "Especially the sample size and analysis plan were difficult to assess due to a lack of standards." – can you clarify what is meant by "and lack of standards" here please?

The recommendations provided are a great addition to this paper and will be of significant benefit to the field if abided by. The 6th recommendation seems like it could be more instructive – what guidelines would the authors recommend based on their findings?

The sample size used was relatively small (admittedly beyond their controls) and from papers published in a single journal. The authors should comment on the limited external validity of their findings in the limitations section.

### ABSTRACT

It will be beneficial for the author to include some of the key outcomes in the abstract (e.g., the exact number of studies with un/disclosed deviations)

I hope the authors find the above comments useful in revising their manuscript for publication.

Sincerely,  
Rob Heirene

## Appendix

R script used to reanalyse data:

```

title: Re-analysis for review
author: "Rob Heirene"
date: "\r format(Sys.time(), '%d %B, %Y')\"
output: html_document

```{r message=FALSE, results = FALSE}
library("dplyr")
```

Remove all objects from work space before starting:
```{r}
rm(list = ls())
# unlink("Workspace_prereg_study_final_analyses_FINAL.RData")
```

```{r}
# Load data:
data<- read.csv("Summary_responses_final.csv") # I extracted just the tab titled "adherence" for
this

# View dataset:
View(data)
names(data)

# Make variable names easier to work with:
data1 <- data %>% rename(none = "Number.of.aspects.with.no.deviations",
  all_disclosed = "Number.of.aspects.with.all.deviations.disclosed",
  undisclosed = "Number.aspects.with.undisclosed.deviations")
```

```

Now I've loaded the data I will check outcomes reported in first paragraph of primary outcome subsection of results:

The authors state here:

- "In our sample, two of the 27 (7%) selected studies did not deviate from the preregistered plan in any of the preregistered methodological aspects (see Figure 1). One study reported all the deviations. In the remaining 24 of 27 (89%) studies, there was at least one item for which a discrepancy between the preregistration plan and the journal article was not fully disclosed. There were no deviations fully disclosed in the article for nine out of 25 (36%) preregistered studies with deviations. In the remaining 16 (64%) studies, at least one deviation was fully disclosed in the article."

```

```{r}
data1 %>% filter(none == "6") # 6 = no deviations at all
# Percentage presented = 7%:
2/27*100

data1 %>% filter(undisclosed > 0) %>% nrow() # anything > 1 indicates an UNdeclared
disclosure
# Percentage presented = 89%:
24/27*100

data1 %>% filter(all_disclosed > 0) %>% nrow() # anything > 1 indicates a declared disclosure

# Percentage presented = 64% (they calcualte out of the 25 with deviations when reporting this):
16/25*100
```

```

Now the second paragraph of this section where the authors state:

- "Non-adherence was most prevalent for the planned sample size, exclusion criteria, and analysis. For six out of 27 (22%) studies, the deviation in sample size is disclosed, as opposed to five (19%) studies, where the deviation is left undisclosed. For five (19%) studies, all changes in exclusion criteria are reported, while in 12 (44%) other studies this is not the case. Finally, in five (19%) studies all deviations in the statistical analysis are reported, but in 13 (48%) not. For this reason, we will discuss some common issues observed for these three aspects."

```

```{r}
# Hypothesis/research question:
table(data$H.RQ)

table(data$Variables)

table(data$Sample.size) # seems right. Percentages:
6/27*100
5/27*100

table(data$Exclusion.criteria) # seems right. Percentages:
5/27*100
12/27*100

table(data$Procedure)

table(data$Analysis) # seems right. Percentages:
5/27*100
13/27*100
```

```

===PREPARING YOUR MANUSCRIPT===

#### ===PREPARING YOUR REVISION IN SCHOLARONE===

- Any electronic supplementary material (ESM).
- If you are requesting a discretionary waiver for the article processing charge, the waiver form must be included at this step.
- If you are providing image files for potential cover images, please upload these at this step, and inform the editorial office you have done so. You must hold the copyright to any image provided.
- A copy of your point-by-point response to referees and Editors. This will expedite the preparation of your proof.

- Ensure that your data access statement meets the requirements at <https://royalsociety.org/journals/authors/author-guidelines/#data>. You should ensure that you cite the dataset in your reference list. If you have deposited data etc in the Dryad repository, please only include the 'For publication' link at this stage. You should remove the 'For review' link.
- If you are requesting an article processing charge waiver, you must select the relevant waiver option (if requesting a discretionary waiver, the form should have been uploaded at Step 3 'File upload' above).
- If you have uploaded ESM files, please ensure you follow the guidance at <https://royalsociety.org/journals/authors/author-guidelines/#supplementary-material> to include a suitable title and informative caption. An example of appropriate titling and captioning may be found at [https://figshare.com/articles/Table\\_S2\\_from\\_Is\\_there\\_a\\_trade-off\\_between\\_peak\\_performance\\_and\\_performance\\_breadth\\_across\\_temperatures\\_for\\_aerobic\\_scope\\_in\\_teleost\\_fishes\\_/3843624](https://figshare.com/articles/Table_S2_from_Is_there_a_trade-off_between_peak_performance_and_performance_breadth_across_temperatures_for_aerobic_scope_in_teleost_fishes_/3843624).

## Author's Response to Decision Letter for (RSOS-211037.R0)

See Appendix A.

## Decision letter (RSOS-211037.R1)

Dear Ms Claesen,

It is a pleasure to accept your manuscript entitled "Comparing Dream to Reality: An Assessment of Adherence of the First Generation of Preregistered Studies" in its current form for publication in Royal Society Open Science. The comments of the reviewer(s) who reviewed your manuscript are included at the foot of this letter.

Please ensure that you send to the editorial office an editable version of your accepted manuscript, and individual files for each figure and table included in your manuscript. You can

send these in a zip folder if more convenient. Failure to provide these files may delay the processing of your proof. You may disregard this request if you have already provided these files to the editorial office.

You can expect to receive a proof of your article in the near future. Please contact the editorial office ([openscience@royalsociety.org](mailto:openscience@royalsociety.org)) and the production office ([openscience\\_proofs@royalsociety.org](mailto:openscience_proofs@royalsociety.org)) to let us know if you are likely to be away from e-mail contact -- if you are going to be away, please nominate a co-author (if available) to manage the proofing process, and ensure they are copied into your email to the journal.

Kind regards,  
Royal Society Open Science Editorial Office  
Royal Society Open Science  
[openscience@royalsociety.org](mailto:openscience@royalsociety.org)

on behalf of Professor Zoltan Dienes (Associate Editor) and Essi Viding (Subject Editor)  
[openscience@royalsociety.org](mailto:openscience@royalsociety.org)

Associate Editor Comments to Author (Professor Zoltan Dienes):  
Associate Editor  
Comments to the Author:

Thank you for your thorough responses to the reviewers' points, which have added important details and clarified previous ambiguities. I hope your paper helps greater integration of pre-registered plans into papers in the future!

## Appendix A

Dear Ms Claesen

On behalf of the Editors, we are pleased to inform you that your Manuscript RSOS-211037 "Comparing Dream to Reality: An Assessment of Adherence of the First Generation of Preregistered Studies" has been accepted for publication in Royal Society Open Science subject to minor revision in accordance with the referees' reports. Please find the referees' comments along with any feedback from the Editors below my signature.

Please submit your revised manuscript and required files (see below) no later than 7 days from today's (ie 17-Aug-2021) date. Note: the ScholarOne system will 'lock' if submission of the revision is attempted 7 or more days after the deadline. If you do not think you will be able to meet this deadline please contact the editorial office immediately.

Kind regards,  
Royal Society Open Science Editorial Office  
Royal Society Open Science  
[openscience@royalsociety.org](mailto:openscience@royalsociety.org)

on behalf of Professor Zoltan Dienes (Associate Editor) and Essi Viding (Subject Editor)  
[openscience@royalsociety.org](mailto:openscience@royalsociety.org)

Associate Editor Comments to Author (Professor Zoltan Dienes):

Associate Editor: 1

Comments to the Author:

I have three excellent and very thoughtful reviews back; all are highly positive about your paper but also providing important points to consider in preparing a revision. I look forward to seeing the revised manuscript.

***Dear Professor Zoltan Dienes and Essi Viding,***

***On behalf of my co-authors, I would like to thank you for the opportunity to revise and resubmit our Manuscript RSOS-211037 "Comparing Dream to Reality: An Assessment of Adherence of the***

***First Generation of Preregistered Studies.*** " I would also like to thank you for extending the deadline for submitting the revised manuscript. The three referees provided excellent and thorough reviews, and this flexibility allowed us to address all remarks carefully.

***In this revision, we implemented two main adjustments. First, we added an extra figure to clarify the steps undertaken in our study. We also adapted Figure 2 (tile plot) by reordering and removing the labels on the x-axis. Second, we adapted the Main outcomes subsection and the Comments from the authors subsection by discussing the results more concisely and systematically.***

***We uploaded two versions of the manuscript, one with and one without tracked changes. You can find our responses to each comment individually below in italic and bold.***

***Thank you once again for your consideration of our revised manuscript.***

***Kind regards,  
Aline Claesen***

Reviewer comments to Author:

Reviewer: 1

Comments to the Author(s)

In this paper, the authors study to what extent the first generation of preregistered studies in psychology (as measured from articles published with the preregistered badge in Psychological Science Feb 2015-Nov 2017) adheres to their preregistration plans and disclose deviations. They start with a sample of 23 articles and 38 preregistration plans, but because of lacking accessibility or lacking minimal methodological detail, the focus is on 16 articles and 27 corresponding articles. This decrease in sample size is very interesting in itself (and could maybe even be mentioned in the abstract?). The main outcome variable is the adherence of the study in the published article to the preregistration plan, as measured by six different items related to e.g. sample size and analysis that were coded as one of three options (no deviations, undisclosed deviation(s), or all deviations disclosed). The other two secondary outcome variables were template use and repository. The results suggest quite some non-adherence and lack of disclosure of deviations, in particular for sample size, exclusion criteria and analysis.

This is a very interesting and important paper. I am extremely positive and just have a few comments that the authors may or may not want to consider.

***Thank you.***

It would be interesting with some more discussion of how severe or problematic the deviations are. I understand that it is not the point of the study to explore whether e.g. non-preregistered control variables that are included lead to  $p < 0.05$  results that otherwise are absent, but would this be possible/are the data for these studies available? If so perhaps that could be a separate study? ***We referred to this point in the Limitations subsection. Besides the fact that the role of deviations is beyond the scope of our study, we also cannot say much more about it, because not all papers in our sample were published with open data.***

For the "Statistical analysis" section on p.7 starting on line 30, it would be interesting with more information on deviations.

***We updated this paragraph.***

When it comes to the recommendations, I am curious about what the authors think about changing the norm of how we write papers to make them more like Registered Reports (even when they are not RRs)? This format would make it easier to show and disclose deviations from preregistration plans etc.

***We added a remark concerning the format of registered reports in Recommendation 3: “For this reason, we suspect that a review-based approach, such as the format of registered reports [12], is superior to the disclosure approach in the sample of our study. However, Hardwicke and Ioannidis [31] discovered some implementation issues in registered reports as well, like non-availability of the plans that are in principle accepted and various ways of registering..”***

I really like the Conclusion section and how the authors acknowledge the efforts of the original authors of the studies they sampled.

***Thank you.***

There are other papers comparing preregistrations with published papers, e.g. Ofose and Posner (2021). I think it would make sense to include a discussion of these papers.

***We included the suggested reference (together with another suggestion from the other reviewers) in the Discussion section: “Our findings are consistent with those from other adherence assessments of preregistered studies. In the fields of economics and political science, Ofose and Posner [28] found in 93 pre-analysis plans registered between 2011 and 2016 that over a third of the papers did not adhere to the planned hypothesis and 18% presented non-preregistered hypothesis tests, which were not disclosed in 82% of the cases. In a more recent study, Heirene et al. [29] reviewed a sample of 20 gambling studies preregistered between 2017 and 2020, and found that 65% contained undisclosed deviations.”***

p.9, line 42: a “d” is missing in “and” in “Finally, an...”.

***We fixed the typo.***

George K. Ofose, Daniel N. Posner (2021). Pre-Analysis Plans: An Early Stocktaking. Accepted, Perspectives on Politics.

Reviewer: 2

Comments to the Author(s)

Dear Authors and Editor,

Thank you for the chance to review this interesting piece of work. The article describes, in a sample of studies that received the Preregistered badge in Psychological Science (2015-2017), the extent of (disclosed and undisclosed) deviations from preregistration plans in the final publication. Corresponding authors of all studies were contacted to obtain comments on the assessments. The authors find that the vast majority of publications included undisclosed deviations, and identify main areas where deviations were more often observed.

This is one of very few articles to assess differences between preregistrations and publication, and I deem it likely very interesting to the readers. Main shortcomings are in my view the intrinsic difficulties of the assessments, and that level of specificity of the preregistered plans is not addressed. Please find below additional comments, which I hope will be of help.

Sincerely,

Cathrine Axfors  
MD, PhD

--

General comments:

1. I agree with the limitation stated that underspecified plans might escape scrutiny and this would be a lamentable development. I would encourage the authors to consider adding an assessment of the level of detail of the preregistered plan. In another study, <https://psyarxiv.com/nj4es>, Heirene et al believed this was relevant when they assessed 53 preregistered plans and so “scored their level of specificity using a 23-item protocol developed to measure the extent to which a clear and exhaustive preregistration plan restricts various researcher degrees of freedom (RDoF; i.e., the many methodological choices available to researchers when collecting and analysing data, and when reporting their findings).”

***The reviewer is absolutely correct that there are varying degrees of specificity in the preregistration plans. However, this was not the aim of our study, so instead we used an absolute cut-off.***

2. The Comments from corresponding authors part is almost handled as a quality control step, while I would encourage the authors to emphasize this step as another data collection. Especially the part about reasons for deviations could perhaps be described a bit more systematically/comprehensively (e.g., categorize the reasons for deviations in a table, give examples, and perhaps to count the number of plans/deviations for which that reason applies).

***We added more structure and more detail to this subsection.***

3. I believe that what is currently described as exclusion criteria are added measurements, which make the paper more interesting. Page 4, Row 15: That many articles were found not accessible or not having enough detail to be assessed for adherence are findings in themselves. I recommend the authors to reformulate so that the study is described as having three parts. In the first (accessibility), there are no exclusion criteria. In the second (minimal detail), there are. In the third (adherence), yet additional ones are added. The third part contains the main outcome. The current grouping into exclusion criteria and main analysis is a bit confusing, especially since the full initial sample (38 preregistration plans) is referred in the secondary outcomes.

Also, with this, I suggest that a flowchart be added to describe the exclusions at different steps. The heading "Exclusions" would be changed to "Accessibility and detail" or a better suited one.

***We slightly adapted the Methods section and included a flowchart to clear up the confusion.***

***Accessibility and minimal detail are exclusion criteria for the adherence assessment, but we did include them for secondary outcomes. We also changed the subtitle from “Excluded preregistration plans” to “Accessibility and minimal detail”, because it’s indeed rather confusing that these studies were not excluded overall.***

4. It would have been of top interest to ascertain whether studies were indeed preregistered. It is salient (although perhaps unsurprising) that the authors deemed this infeasible. For clinical trials, it is often stated in registrations when the first patient was enrolled. I encourage the authors to add a sentence or two in the discussion about this and any thoughts on how such ascertainment could be implemented for observational studies (at other some stage of the publication process, if not now).

***To the first recommendation, we added: “Further, the time stamp is more interpretable, if authors report when data collection started.”***

Specific comments:

5. Could the circumstances around badges be explained briefly? Are all the studies in the sample performing confirmatory hypothesis testing? I assume so, but it would be relevant to state. How were changes from plans to publication categorized that were explicitly described as exploratory (if such were observed)?

***We included extra information on the badge in Sample subsection: “The Preregistered badge indicated the presence of a preregistration plan. In order to earn the badge, authors had to fill out an open practices disclosure document, in which they declared that there is a permanent path to a preregistration plan on an online open access repository, and in which they could disclose deviations if any. The preregistration plans were not reviewed.”***

***Any type of study can be preregistered. Because we wanted comparability between the studies, we employed minimal detail as an exclusion criterion to select studies that were preregistered in a similar way (and that conducted confirmatory hypothesis testing). We added the following sentence: “That is, the purpose was to select studies of which adherence can be evaluated regarding six methodological items that indicated confirmatory hypothesis testing, the minimal detail criterion does not constitute an evaluation of the quality of the preregistration plan.” In the measures and outcomes subsection, we also included “Parts of the papers that were clearly labelled as exploratory were not included in the comparison (i.e., this was coded as no deviation rather than all deviations disclosed).”***

6. Page 4, Row 57 onwards. All exclusion criteria and “primary measures” were assessed by two independent raters. What are the primary measures referred here? If referring to main outcome, the last sentence on the same page indicates the opposite: “Adherence was assessed by one rater”, i.e., not independently by two raters. What does it mean that adherence was assessed “under the supervision” of two co-authors? Asking since adherence is the main outcome, and as the authors write, it proved to be a far from trivial task.

***We adapted the Procedure subsection, because it was indeed a bit confusing. We now clarify that two raters independently assessed accessibility, minimal detail and adherence. However, in a later stage, we introduced some changes to the coding scheme for adherence (which are also explicated in the Disclosure subsection). The assessment was then updated to the new coding scheme (8 to 6 items) by one rater, in consultation with the last two authors if a change was deemed necessary.***

7. Page 5, row 34. “Did not include a planned sample size”. Were there any studies without a planned sample size, with a rationale for why the sample size was not prespecified?

***Yes, two corresponding authors provided a rationale after we contacted them. We refer to their rationale in the Comments from corresponding authors subsection: “In particular, for the minimal detail criterion, two corresponding authors pointed out that they did not preregister certain study details because of the nature of their study. One conducted a direct replication and indicated that study details can be retrieved from the original study. Another corresponding author clarified that it is not realistic to set the sample size beforehand for an observational study.”***

8. Page 5, row 38. Here, like in some other parts of the manuscript, authors add their observations and experiences as they go. While I can recognize the need to disclose such details from the perspective of being an author, as a reader I do not always appreciate the side details before having understood the main picture. I would encourage the authors to refrain from interpreting the findings before the discussion section, and to place for example the content under “preliminary observations” in the limitations section of the discussion.

***We agree that usually such details belong to the Discussion and not the Results section. However, here the qualitative remarks are an important result. They are not merely a limitation, they show that preregistration is not necessarily transparent in itself. For this reason, we prefer not to move this entire part to the limitations. However, we do refer to it in the first limitation.***

***Based on the reviewer’s suggestion, we added following sentence to the Discussion section: “Our preliminary observations also showed that even with the preregistration plan at hand, it is challenging to distinguish what was planned a priori from what was not.”***

9. Page 5, last row, “there were no deviations fully disclosed in the article for nine out of 25...”

Suggest to reformulate to “None of the deviations were fully disclosed in the article for nine out of 25

preregistered studies with deviations”, for readability. And, “In the remaining 16 ... studies, at least one [of the] deviations was fully disclosed in the article.”

**We implemented the reviewer’s suggestion.**

10. Figure 1. Layout and editing. The study names are a bit confusing for readers who don’t revisit the assessment data file, but this is a minor point.

**We removed the study labels, because they are indeed only informative for readers who consult the assessment data file (and there they can find the same information as in the figure).**

11. To add to methods section: How are “variables” defined and what constitutes deviations from “variables”? Since the adherence criteria didn’t specify operationalization of variables. Is it the number of variables, or their description? Generally, the description on page 4, row 30, of adherence items lacks details.

12. Probably related: What did the authors mean with that a variable is operationalized? “Operationalization of the variables” was required for inclusion in the adherence analysis, but that item seems to be baked into the “variables” item in the very adherence analysis.

**We included the following in the Measures and outcomes subsection: “Note that the adherence items somewhat differed from the minimal detail items. In particular, we selected studies that listed variables and their operationalization (i.e., what and how the variables would measure or control). Due to frequent changes in terminology, we sometimes had to identify variables based on their description. Therefore, the variables item in the adherence assessment covers operationalization as well. Also note that we did not require exclusion criteria for minimal detail, but did include this item in the adherence assessment. If no exclusion criteria were reported in the paper, and no exclusion criteria were preregistered, then there was no deviation.”**

13. It would add strength to the manuscript if the authors discussed their findings in relation to similar studies. From Wikipedia, heading Preregistration (science), there seem to exist at least one or two, “... researchers rarely follow the exact research methods and analyses that they preregister (Abrams et al., 2020; Claesen et al., 2019; Heirene et al., 2021; see also Boghdadly et al., 2018; Singh et al., 2021; Sun et al., 2019)”. Some of these are for RCTs, but comparisons can still be relevant. Also, a comment on how the methods here differed from the Heirene et al study (and limitations/strengths in light of that) would be excellent in my opinion.

**In the introduction, we already included references to studies on RCT registrations. In this revision, we included references to the studies by Ofosu and Posner (2021) and Heirene (2021) in the discussion in this paragraph: “Our findings are consistent with those from other adherence assessments of preregistered studies. In the fields of economics and political science, Ofosu and Posner [27] found in 93 pre-analysis plans registered between 2011 and 2016 that over a third of the papers did not adhere to the planned hypothesis and 18% contained non-preregistered hypothesis tests, which were not disclosed in 82% of the cases. In a more recent study, Heirene et al. [28] reviewed a sample of 20 gambling studies preregistered between 2017 and 2020, and found that 65% contained undisclosed deviations.” We also refer to Heirene et al. in the Limitations subsection: “That the comparison between plan and paper is challenging, is also evident in the study by Heirene et al. [28]. For 40.6% of the scores, they could not determine adherence due to a lack of information in the preregistration plan, or both the plan and the paper,” and “The study by Heirene et al. supports this conjecture [29]. They found an increase in specificity and adherence over time in preregistered studies from 2017 to 2020. However, they argue that there is still room for improvement.”**

14. About generalizability. Could the authors add in the discussion how many articles have received the preregistered badge during the years after 2017? Just to give the reader a sense of the development and how big a portion is assessed here.

***There are no numbers available on how many articles have received the preregistered badge. As we discuss in the limitations, the generalizability of this study is rather limited, due to the small (and early) sample from one journal.***

15. It is a challenge to strike the right balance of specificity and sensitivity in the definition of deviations. For sample size, the authors comment that it may be trivial with a small deviation from the planned number, a view endorsed by some corresponding authors which I sympathize with. For details that were “compatible” with the plan but represented an added level of granularity, perhaps these escaped the check too lightly (see comment 1). Back to sample size: just because this is a number (and so a difference is easier to detect), it’s not sure that it’s a relevant difference, much like the case with statistically significant results in a very large dataset. Do the authors believe that a priori sample size specification is relevant for all the articles assessed? If not, perhaps that item should be reconsidered to allow some percentage change as no deviation.

***We agree that finding the right balance of specificity and sensitivity is challenging, and that there are multiple ways to define deviations. We highlighted the importance of general guidelines for reporting preregistered studies in recommendation 7. As we also mentioned in the Limitations subsection, our approach only focused on reporting and not on the impact of the deviations. In the Conclusion section, we pointed out that we do not claim that the deviations observed by us constitute evidence for questionable research practices.***

***It is true that it is easier to detect a difference in a number, but it is also easier to disclose this difference. This does not hold for the statistical analysis, for instance, because there are many more decisions involved, and thus more deviations possible.***

16. Personal preference: parts of the results that describe methodology (how sample size, exclusion criteria, and statistical analysis were operationalized) I would prefer to see in the methods section. At present, the description of the results mixes methods details that have previously not been disclosed, and interpretations, with the results reporting. I admit that differences may exist between research fields, and I come from a tradition with more separated sections where concise is key.

***We adapted the results section.***

17. The authors have done great work. I was surprised by the at times defensive tone, and that the authors seem to have felt the need to excuse their definitions or entire endeavour. Then I read the comments from some of the contacted corresponding authors, which are in my view quite unforgiving. Most readers will probably agree that this type of analysis is a necessary next step after introducing preregistration as a phenomenon, and I don’t believe it is the responsibility of the authors to bolster all reactions.

***Thank you. We have opted for a more humble approach, because these authors are part of the first group of researchers that preregistered their studies. There was little guidance back then, and we wish to avoid to accuse the corresponding authors of QRP’s.***

Reviewer: 3

Comments to the Author(s)

Thursday, 5 August 2021

#### OVERALL COMMENTS

Thank you for the opportunity to review this interesting manuscript. The findings presented advance the field of psychology and beyond, the conclusions are supported by the data, and the methods appear scientifically sound. There are several concerns that I have (e.g., the failure to contextualise this study and the outcomes with the existing literature) but the authors should be able to address these in a revision. Below I provide specific comments on each of the individual sections in the

manuscript.

## INTRODUCTION

### FIRST PARAGRAPH

The authors state: “During the research process, researchers inevitably make numerous decisions, collectively known as researcher degrees of freedom” — please provide examples of such decisions for researchers less familiar with the concept of researcher degrees of freedom

***We followed the reviewer’s suggestion and added some examples:***

***“During the research process, researchers inevitably make numerous decisions, collectively known as researcher degrees of freedom [1]. Among others, researchers need to decide on the number of participants, possible data transformations, the treatment of outlying data, the inclusion of covariates, and so on. When this flexibility is exploited, the probability of a type I error is drastically increased [2,3], or the effect size can be inflated. A common example is the practice of optional stopping, which involves stopping data collection based on interim data analysis, in order to reach the desired result.”***

The authors state: “One way to increase research transparency is to preregister research, which involves freezing the analytic choices on a public third-party repository prior to seeing, or ideally prior to collecting, the data [9]. By specifying decisions before data collection, researcher degrees of freedom are restricted, and decisions that are made during the data collection and analysis cannot be mistakenly reported as a priori.” — for the purpose of clarity it would be worth noting that researchers often preregister more than just their analytical decisions (e.g., hypotheses, study design etc.), even if these elements are not the focus of this paper.

***Indeed, we have replaced “analytic choices” with “decisions regarding the study (e.g., study design, data collection and data analysis)”. In the abstract we replaced it with “research decisions”.***

### SECOND PARAGRAPH

There is a study that has looked at adherence to preregistrations in the political science and economics literature by Ofsu & Posner (2019) — Ofsu, G., & Posner, D. N. (2019). Pre-analysis Plans: A Stocktaking (Pre-Print). MetaArXiv. <https://doi.org/10.31222/osf.io/e4pum>

It would be helpful if the authors could briefly discuss this study here to better contextualise their study and describe the current state of existing research on this topic (or lack thereof)

Similarly, my colleagues and I have just pre-printed a study on this topic which is highly relevant (although this was very recent and therefore the authors will not have had time to incorporate this into their article). — Heirene, R., LaPlante, D., Louderback, E. R., Keen, B., Bakker, M., Serafimovska, A., & Gainsbury, S. M. (2021, July 16). Preregistration specificity & adherence: A review of preregistered gambling studies & cross-disciplinary comparison.

<https://doi.org/10.31234/osf.io/nj4es>

***We included following paragraph in the Discussion section: “Our findings are consistent with those from other adherence assessments of preregistered studies. In the fields of economics and political science, Ofsu and Posner [27] found in 93 pre-analysis plans registered between 2011 and 2016 that over a third of the papers did not adhere to the planned hypothesis and 18% contained non-preregistered hypothesis tests, which were not disclosed in 82% of the cases. In a more recent study, Heirene et al. [28] reviewed a sample of 20 gambling studies preregistered between 2017 and 2020, and found that 65% contained undisclosed deviations.”***

### FINAL PARAGRAPH

It would be helpful if the authors more clearly stated the aim of the study towards the end of this paragraph (if space is an issue, then the sentences describing the findings/ conclusions can be removed, but this is more of a personal preference than a genuine issue).

***We have slightly adapted the final paragraph:***

***“In the current study, we focus on deviations from the preregistration plan in pioneer research articles with a Preregistered badge in Psychological Science, and whether these deviations are disclosed. We identify possible explanations for deviations and formulate several recommendations to increase transparency in preregistered studies in future research.”***

#### METHODS

There is no information on data analysis here — I recognise that the authors only include summary statistics and don't report the results of any inferential statistical tests, but it would still be useful for the authors to state what software+packages they used to analyse the data and develop figures and where we can access the analysis code (if available?)

***Indeed, because we limited the results to summary statistics, we did not include these details. However, following the reviewer's suggestion, we included it: “Our coding scheme and assessment can be found here: [https://osf.io/f49an/?view\\_only=09b2d9c6ab1d4f6f973461fe151243bc](https://osf.io/f49an/?view_only=09b2d9c6ab1d4f6f973461fe151243bc). Analyses are conducted in R (version 4.0.2), employing the following R packages: *xlsx* (version 0.6.5), *tidyverse* (version 1.3.0), *ggplot2* (version 3.3.2) [21–24]. R script for main outcomes and Figures 2 and 3 are available here: [https://osf.io/bs5q2/?view\\_only=d81b47d8f9eb45b7924153529f2f1a53](https://osf.io/bs5q2/?view_only=d81b47d8f9eb45b7924153529f2f1a53).”***

#### DISCLOSURE SUBSECTION

While the reasons stated by the authors provide a reasonable rationale for why the study was not preregistered, it may appear odd to those who are less familiar with the process of preregistration (or who are critical of it) that the authors did not preregister their own study on preregistration practices. Perhaps this could be avoided by discussing the purpose and use of preregistration to a greater extent in the introductory section (e.g., use in confirmatory/hypothesis testing research vs. exploratory studies).

***We included the following sentence: “Therefore, the practice is most suitable for confirmatory research, but does not favour it. It merely distinguishes confirmatory from exploratory results [5].”***

Can the authors clarify what is meant by “open practices disclosure items” please?

***We added the following footnote: “Every article that was published in Psychological Science with one or more open practices badges was accompanied by an open practices disclosure document, containing five disclosure items adopted from the guidelines of the OSF (<https://osf.io/tvyxz/wiki/2.%20Awarding%20Badges/>).”***

I can't access the link to the assessment procedure. I have requested access to this from the authors and would like to review it before finalising my review. Is there any reason why this assessment is not available to the public? Especially given that the study has been available for some time as a preprint?

***The assessment will be public upon publication the manuscript. Readers can request access for now. (I noticed the reviewer did, I sent an email but perhaps it ended up in the spam folder).***

#### SAMPLE SUBSECTION

Can the authors better describe the characteristics of the preregistrations and studies included, please? For example, how many studies were published in each of the three years studied (2015, 2016, 2017)? What types of studies were included and did the authors exclude systematic reviews and RCTs (which have different registration requirements)? Where were the studies preregistered (OSF, PROSPERO, aspredicted)? What preregistration templates were used? Both research by myself and colleagues and Bakker et al. (2021; doi:10.1371/journal.pbio.3000937) have shown the templates used affect the quality of the preregistration? I recognise that the authors were not looking at the quality of the preregistration itself, but it could still be interesting from an adherence perspective. Perhaps a table could be used to display most of this information concisely? (ADDITION: I see the authors describe some of these variables in the secondary outcomes; still, perhaps this

information could be more easily summarised as a table?)

***In text, we included the number of preregistered papers per year. We included a flowchart with a summary of information on the preregistration plans (Figure 1).***

#### MEASURES AND OUTCOMES SUBSECTION

It's great that the authors have described their process of screening papers/preregistrations very clearly at the beginning of this section.

Minor issue: the authors jump from past tense to present tense (this sentence "Without full disclosure, deviations are undisclosed.") then back to past tense again in the second paragraph of this section.

***Thank you. We changed this sentence, but we corrected other verb tenses based on the reviewer's comment.***

#### PROCEDURE SUBSECTION

Having just performed a very similar study, I find it concerning that only one researcher scored adherence to preregistrations. In our study, two researchers independently scored preregistration adherence and found it extremely difficult and time-consuming to make these comparisons — we spent almost as much time reviewing and discussing our scoring as we did actually doing the scoring due to ambiguities and inconsistencies between preregistrations and papers in terminology, phrasing, structure etc. and clearly the rater faced similar difficulties based on the statements in the second paragraph of this section. Can the authors provide a bit more information about they avoided errors in this process and how the sole rater was supervised (e.g., were any of the scores directly checked by another author)?

***We had the same experience: it is extremely difficult to compare the preregistration to the article. We also spent more time on reviewing the scoring than the scoring itself. There were two independent raters: the first and second author. Later, we updated the coding scheme. This was done by the first author (the second left the research group at the time and was on maternal leave). Every code she wished to change was discussed with the two last authors. We updated the Procedure subsection and hope that our description of the procedure is no longer confusing.***

The authors state: "As a result, there were cases in which it was difficult to assess whether there was a deviation from the preregistration plan. Whenever there was reasonable doubt about a deviation from the preregistration plan, this was coded as no deviations " – were the number of instances when this occurred and recorded? I think if the team could not adequately tell whether authors of their sample papers had deviated or not from their registration then this is a separate and noteworthy phenomenon. As the authors discuss later, their scoring process also rewards poor specification by not separating out these cases. Not differentiating these instances concerns me as when scoring adherence on 20+ items in our paper, nearly 41% of the scores we gave were classified "Unable to tell [due to a lack of detail in the pre-registration, paper, or both]"(<https://psyarxiv.com/nj4es/>). Admittedly, this figure is likely to be lower in the present study as the authors required preregistrations to contain a minimal set of details (we did not) that were then compared.

***It is very informative to include this score, however, the number of instances were not recorded. We eventually always came to an agreement, and we found it difficult to define when this would not be possible, because the authors almost never reported the study in the exact same way as it was preregistered. We mainly focused on consistency: If the authors preregistered to perform a regression analysis and they reported a multilevel regression analysis, this is consistent. In the Limitations subsection we included: "That the comparison between plan and paper is challenging, is also evident in the study by Heirene et al. [28]. For 40.6% of the scores, they could not determine adherence due to a lack of information in the preregistration plan, or both the plan and the paper."***

## RESULTS

### PRELIMINARY OBSERVATIONS SUBSECTION

This section is an important addition and mirrors many of the issues we faced when making these comparisons. This qualitative/process related detail can be lost in this sort of study and so I was very pleased to see the authors include this here in the findings.

***Thank you.***

### PRIMARY OUTCOME: ADHERENCE SUBSECTION

I finally got access to the assessment procedures and raw data from the link included in Figure 1's note. I downloaded and reanalysed the data to reproduce the outcomes reported in this subsection. I successfully reproduced all outcomes.

I have included the code used for this below (appendix) and I attached the full script & html output as documents to this review.

***Thank you for taking the time to reproduce our outcomes!***

I find the X axis labels on Figure 1 to be very confusing. Can the authors try and make this more readable without having to look at the raw data (although even then I find it confusing)? Just changing the labels to "paper 21 (2 studies)" would be easier to understand.

***We removed the study labels, because even if we would change them, they still only would make sense upon consulting the assessment file.***

### SAMPLE SIZE SUBSECTION

It seems like there is a lot of information reported here for little gain. The authors could much more concisely say that they treated any discrepancy between the exact sample size preregistered and the sample size reported in the paper as a deviation, without the need to provide a lengthy example to illustrate this.

It would be useful if the office can provide some more qualitative summary information here rather than detailed qualitative examples. For instance, what was the average/median/range for the discrepancy between preregistered and actual sample size? Was there any pattern in the studies that disclosed the deviation (e.g., smaller discrepancies?). Overall, it seemed more like I was reading a discussion+recommendations section here rather than a description of the results.

### Statistical analysis subsection

Same issues as stated above apply here. For example, the following statement appears odd in a results section and should appear in the discussion: "Detailing the statistical analysis is probably the hardest part of writing a preregistration plan. It is, therefore, no surprise that especially for the statistical analysis, deviations are likely to occur."

Why don't the authors provide a subsection for the other 3 areas of adherence studied (i.e., variables, hypotheses, and procedures)? I recognise these areas had the most overall deviations, but there were as many undeclared deviations relating to hypotheses as there were relating to sample sizes

### EXCLUSION CRITERIA SUBSECTION

Again, there is a very lengthy quotation from one of the papers included here that could be cut down substantially. Again, there is a similar feel to this paragraph like it is more of a discussion section or form a conceptual paper, rather than being a description of the results.

***We adapted these three subsections in the results section.***

### COMMENTS FROM CORRESPONDING AUTHORS

This is another interesting addition to this paper, and I commend the authors for contacting all authors in their sample.

**Thank you.**

## DISCUSSION

A comparison of the findings presented here with those from the existing literature in this area would be beneficial here before moving on to offer recommendations, especially as later in this authors state: “Time is a very plausible game changer, and it is likely that authors, reviewers, and editors are improving how they deal with preregistration over time. New assessments might reveal an improvement compared to what we found in this sample. However, this assessment is important to reflect on current and new tools for preregistration.” — more recent assessments are now available.

**Thank you for pointing this out. We included a reference to the reviewer’s study: “The study by Heirene et al. supports this conjecture [30]. They found an increase in specificity and adherence over time in preregistered studies from 2017 to 2020. However, they argue that there is still room for improvement. Assessments like the current are important to reflect on present and future tools for preregistration.”**

The authors state: “Especially the sample size and analysis plan were difficult to assess due to a lack of standards.” — can you clarify what is meant by “and lack of standards” here please?

The recommendations provided are a great addition to this paper and will be of significant benefit to the field if abided by. The 6th recommendation seems like it could be more instructive — what guidelines would the authors recommend based on their findings?

**We added following sentence: “For example, should researchers report that they deviated from the sample size when they only sampled a few participants less or more than planned? If the statistical model was reported in more detail in the paper in the preregistration plan, should researchers report these details as deviations?”**

The sample size used was relatively small (admittedly beyond their controls) and from papers published in a single journal. The authors should comment on the limited external validity of their findings in the limitations section.

**Indeed, we made comment on this limitation more explicit: “Finally, and perhaps most importantly, it should be kept in mind that our sample consists of a relatively small number of preregistered studies of the first generation from a single journal, and the generalizability of our findings is limited. ”**

## ABSTRACT

It will be beneficial for the author to include some of the key outcomes in the abstract (e.g., the exact number of studies with un/disclosed deviations)

**We updated the abstract and included the main outcomes.**

I hope the authors find the above comments useful in revising their manuscript for publication.

Sincerely,

Rob Heirene

## Appendix

R script used to reanalyse data:

```

title: Re-analysis for review
author: "Rob Heirene"
date: "`r format(Sys.time(), '%d %B, %Y')`"
output: html_document

```{r message=FALSE, results = FALSE}
library("dplyr")
```

Remove all objects from work space before starting:
```{r}
rm(list = ls())
# unlink("Workspace_prereg_study_final_analyses_FINAL.RData")
```

```{r}
# Load data:
data<- read.csv("Summary_responses_final.csv") # I extracted just the tab titled "adherence" for this

# View dataset:
View(data)
names(data)

# Make variable names easier to work with:
data1 <- data %>% rename(none = "Number.of.aspects.with.no.deviations",
  all_disclosed = "Number.of.aspects.with.all.deviations.disclosed",
  undisclosed = "Number.aspects.with.undisclosed.deviations")
```
```

Now I've loaded the data I will check outcomes reported in first paragraph of primary outcome subsection of results:

The authors state here:

- "In our sample, two of the 27 (7%) selected studies did not deviate from the preregistered plan in any of the preregistered methodological aspects (see Figure 1). One study reported all the deviations. In the remaining 24 of 27 (89%) studies, there was at least one item for which a discrepancy between the preregistration plan and the journal article was not fully disclosed. There

were no deviations fully disclosed in the article for nine out of 25 (36%) preregistered studies with deviations. In the remaining 16 (64%) studies, at least one deviation was fully disclosed in the article."

```
```{r}
data1 %>% filter(none == "6") # 6 = no deviations at all
# Percentage presented = 7%:
2/27*100

data1 %>% filter(undisclosed > 0) %>% nrow() # anything > 1 indicates an UNdeclared disclosure
# Percentage presented = 89%:
24/27*100

data1 %>% filter(all_disclosed > 0) %>% nrow() # anything > 1 indicates a declared disclosure

# Percentage presented = 64% (they calculated out of the 25 with deviations when reporting this):
16/25*100
```
```

Now the second paragraph of this section where the authors state:

- "Non-adherence was most prevalent for the planned sample size, exclusion criteria, and analysis. For six out of 27 (22%) studies, the deviation in sample size is disclosed, as opposed to five (19%) studies, where the deviation is left undisclosed. For five (19%) studies, all changes in exclusion criteria are reported, while in 12 (44%) other studies this is not the case. Finally, in five (19%) studies all deviations in the statistical analysis are reported, but in 13 (48%) not. For this reason, we will discuss some common issues observed for these three aspects."

```
```{r}
# Hypothesis/research question:
table(data$H.RQ)

table(data$Variables)

table(data$Sample.size) # seems right. Percentages:
6/27*100
5/27*100

table(data$Exclusion.criteria) # seems right. Percentages:
5/27*100
12/27*100

table(data$Procedure)

table(data$Analysis) # seems right. Percentages:
5/27*100
13/27*100
```
```

===PREPARING YOUR MANUSCRIPT===

- one version identifying all the changes that have been made (for instance, in coloured highlight, in bold text, or tracked changes);
- a 'clean' version of the new manuscript that incorporates the changes made, but does not highlight them. This version will be used for typesetting.

===PREPARING YOUR REVISION IN SCHOLARONE===

- An individual file of each figure (EPS or print-quality PDF preferred [either format should be produced directly from original creation package], or original software format).
  - An editable file of each table (.doc, .docx, .xls, .xlsx, or .csv).
  - An editable file of all figure and table captions.
- Note: you may upload the figure, table, and caption files in a single Zip folder.
- Any electronic supplementary material (ESM).
  - If you are requesting a discretionary waiver for the article processing charge, the waiver form must be included at this step.
  - If you are providing image files for potential cover images, please upload these at this step, and inform the editorial office you have done so. You must hold the copyright to any image provided.
  - A copy of your point-by-point response to referees and Editors. This will expedite the preparation of your proof.

- Ensure that your data access statement meets the requirements at <https://royalsociety.org/journals/authors/author-guidelines/#data>. You should ensure that you cite the dataset in your reference list. If you have deposited data etc in the Dryad repository, please only include the 'For publication' link at this stage. You should remove the 'For review' link.
- If you are requesting an article processing charge waiver, you must select the relevant waiver option (if requesting a discretionary waiver, the form should have been uploaded at Step 3 'File upload' above).
- If you have uploaded ESM files, please ensure you follow the guidance at <https://royalsociety.org/journals/authors/author-guidelines/#supplementary-material> to include a suitable title and informative caption. An example of appropriate titling and captioning may be found at [https://figshare.com/articles/Table\\_S2\\_from\\_Is\\_there\\_a\\_trade-off\\_between\\_peak\\_performance\\_and\\_performance\\_breadth\\_across\\_temperatures\\_for\\_aerobic\\_scope\\_in\\_teleost\\_fishes\\_/3843624](https://figshare.com/articles/Table_S2_from_Is_there_a_trade-off_between_peak_performance_and_performance_breadth_across_temperatures_for_aerobic_scope_in_teleost_fishes_/3843624).

\*\*\*\*\*

Journal Name: Royal Society Open Science

Journal Code: RSOS

Online ISSN: 2054-5703

Journal Admin Email: [openscience@royalsociety.org](mailto:openscience@royalsociety.org)

Journal Editor: Andrew Dunn

Journal Editor Email: [openscience@royalsociety.org](mailto:openscience@royalsociety.org)

MS Reference Number: RSOS-211037

Article Status: SUBMITTED

MS Dryad ID: RSOS-211037

MS Title: Comparing Dream to Reality: An Assessment of Adherence of the First Generation of Preregistered Studies

MS Authors: Claesen, Aline; Gomes, Sara; Tuerlinckx, Francis; Vanpaemel, Wolf

Contact Author: Aline Claesen

Contact Author Email: [aline.claesen@kuleuven.be](mailto:aline.claesen@kuleuven.be)

Contact Author Address 1:

Contact Author Address 2:

Contact Author Address 3:

Contact Author City: Leuven

Contact Author State:

Contact Author Country: Belgium

Contact Author ZIP/Postal Code: 3000

Keywords: Preregistration, Researcher Degrees of Freedom, Open Science, Psychological Science, Transparency

Abstract: Preregistration is a method to increase research transparency by documenting analytic choices on a public, third-party repository prior to any influence by data. It is becoming increasingly popular in all subfields of psychology and beyond. Adherence to the preregistration plan may not always be feasible, and even is not necessarily desirable, but without disclosure of deviations, readers who do not carefully consult the preregistration plan might get the incorrect impression that the study was exactly conducted and reported as planned. In this paper, we have investigated adherence and disclosure of deviations for the first generation of preregistrations. For all articles published with the Preregistered badge in Psychological Science between February 2015 and November 2017, we assessed the adherence to their corresponding preregistration plans and the disclosure of deviations. We shared our assessments with the authors of the preregistered studies and updated our assessment based on their feedback. We mainly observed (un)disclosed deviations from the plan regarding the sample size, exclusion criteria, and statistical analysis. This closer look at preregistrations of the first generation reveals possible hurdles for reporting preregistered studies and provides input for future reporting guidelines. We discuss examples and possible explanations and, provide recommendations for preregistered research.

EndDryadContent